# Changes of Arbuscular Mycorrhizal Fungal Community and Glomalin in the Rhizosphere along the Distribution Gradient of Zonal *Stipa* Populations across the Arid and Semiarid Steppe

Xiaodan Ma,[a,b] Jingpeng Li,[a,b] Fucheng Ding,[a,b] Yaxin Zheng,[a,b] Lumeng Chao,[a,b] Haijing Liu,[a,b] Xinyan Liu,[a,b] Hanting Qu,[a,b] Yuying Bao[a,b]

[a]Key Laboratory of Forage and Endemic Crop Biotechnology, Ministry of Education, School of Life Sciences, Inner Mongolia University, Hohhot, People's Republic of China
[b]State Key Laboratory of Reproductive Regulation and Breeding of Grassland Livestock, Inner Mongolia University, Hohhot, People's Republic of China

**ABSTRACT** Arbuscular mycorrhizal fungi (AMF) have been reported to have a wide distribution in terrestrial ecosystems and to play a vital role in ecosystem functioning and symbiosis with *Stipa* grasses. However, exactly how AMF communities in the rhizosphere change and are distributed along different *Stipa* population with substituted distribution and their relationships remain unclear. Here, the changes and distribution of the rhizosphere AMF communities and their associations between hosts and the dynamic differences in the glomalin-related soil protein (GRSP) in the rhizosphere soil of seven *Stipa* species with spatial substitution distribution characteristics in arid and semiarid grasslands were investigated. Along with the substituted distribution of the *Stipa* populations, the community structures, taxa, species numbers, and alpha diversity index values of AMF in the rhizosphere changed. Some AMF taxa appeared only in certain *Stipa* species, but there was no obvious AMF taxon turnover. When the *Stipa baicalensis* population was replaced by the *Stipa gobica* population, the GRSP tended to decline, whereas the carbon contribution of the GRSP tended to increase. *Stipa grandis* and *Stipa krylovii* had a great degree of network modularity of the rhizosphere AMF community and exhibited a simple and unstable network structure, while the networks of *Stipa breviflora* were complex, compact, and highly stable. Furthermore, with the succession of zonal populations, the plant species, vegetation coverage, and climate gradient facilitated the differentiation of AMF community structures and quantities in the rhizospheres of different *Stipa* species. These findings present novel insights into ecosystem functioning and dynamics correlated with changing environments.

**IMPORTANCE** This study fills a gap in our understanding of the soil arbuscular mycorrhizal fungal community distribution, community composition changes, and diversity of *Stipa* species along different *Stipa* population substitution distributions and of their adaptive relationships; furthermore, the differences in the glomalin-related soil protein (GRSP) contents in the rhizospheres of different *Stipa* species and GRSP's contribution to the grassland organic carbon pool were investigated. These findings provide a theoretical basis for the protection and utilization of regional biodiversity resources and sustainable ecosystem development.

**KEYWORDS** AMF communities, GRSP, *Stipa* taxa, changes, rhizosphere

Address correspondence to Yuying Bao, ndbyy@imu.edu.cn.

The authors declare no conflict of interest.

Arid and semiarid grasslands have attracted wide attention, arising from the declining forage species, soil desertification, and modifications to soil microbial communities due to global climate change (e.g., severe and frequent drought and heavy rainfall events) and anthropogenic activities (e.g., grazing) (1–4). Arid and semiarid grasslands in northern China have a horizontal zonal *Stipa* grassland community composed of *Stipa* as constructive species, due to their significant hydrothermal gradients (5). *Stipa* plants also act as an important

material basis for grassland animal husbandry and can maintain grassland balance and a good ecological environment (6). Arbuscular mycorrhizal fungi (AMF) are important underground organisms that can form mutualistic symbioses with a majority of the plants in grassland ecosystems (7–9). As highlighted by increasing evidence, AMF maintain vegetation cover, productivity, diversity, and soil nutrient resource use efficiency in arid and semiarid areas, while facilitating ecosystem succession, to a certain extent, by adjusting plant interspecific relations (10–13). Moreover, AMF can improve soil stability and erosion resistance in arid and semiarid areas by giving rise to the glomalin-related soil protein (GRSP) (14, 15). Furthermore, the GRSP is believed to be a vital source and component of the organic carbon pool in barren environments (16, 17), and the response of AMF to climate change is expected to be a critical factor in predicting changes in community structure and function in ecosystems (18, 19). Thus, whether the AMF community also exhibits zonal distribution characteristics or whether the community structure of AMF changes along the alternative distribution of aboveground vegetation and the response of the GRSP contents to changes in arid and semiarid environments remain unclear.

In terrestrial ecosystems, plants and soil exhibit obvious zonal distribution characteristics corresponding to climate types (20, 21), whereas aboveground vegetation generally has a significant effect on the AMF community in the soil (22). Numerous studies focusing on the AMF communities in natural ecosystems reported that these communities are shaped by multiple processes, including external biotic forces (e.g., plant community, host plant species, and interspecific interactions) (23) and abiotic filtering (edaphic, climatic, and geographical factors) (24, 25), as well as dispersal and stochasticity (26). According to several previous studies, edaphic factors, especially soil pH and nutrient availability, and climate parameters have been proposed as drivers of the spatial diversity and structure of rhizosphere AMF communities and their distribution patterns (23, 27, 28). Moreover, although the correlation between the host plants and AMF taxa is generally considered weakly specific (23, 29), the community composition changes and geographical distribution of AMF in rhizosphere soil are still driven by changes in the plant species (30–32). The rhizosphere AMF community structure and spatial distribution and related influencing factors of many vital plants in the above-described system have been systematically studied for extended periods by many scholars (16, 33–37). Specifically, the prior studies of the rhizosphere soil of *Stipa* species are still fragmented due to small-scale geographic scope and lack systematic data from natural ecosystems (7, 23, 38). Thus, it is necessary to investigate the rhizosphere AMF communities along the alternative distribution of *Stipa* species on a large geographic scale for the effects of an enormously wide variety of ecological factors on them.

*Stipa* grassland is one representative of the temperate grasslands unique to the Eurasian steppe (6). The genus *Stipa* is a grazing-resistant, drought-resistant, mycorrhizal, and perennial bunchgrass with sand sheaths, which can help plants tolerate abiotic stresses (6, 39). The north-south heat difference and the east-west southeast monsoon wind system differentiate meadow grassland (*Stipa baicalensis* population), typical grassland (*Stipa grandis* and *Stipa krylovii* populations), and desert grassland (*Stipa breviflora*, *Stipa gobica*, *Stipa klemenzii* and *Stipa glareosa* populations) in arid and semiarid grasslands (5, 6, 40). The study transect provides a suitable area and unique opportunity for this investigation due to the high heterogeneity of abiotic factors and geographic substituted distribution patterns of unique *Stipa* species (Fig. 1) (6, 14). Accordingly, in this study, the synergetic distribution and variations in the GRSP, populations, and community composition of AMF in *Stipa* rhizosphere soils with the horizontal zonal distribution of the *Stipa* populations, as well as the response to an enormously wide variety of ecological factors under this natural ecosystem, were deeply clarified to provide clues for predicting the symbiotic occurrence and evolution of grassland ecosystems. This study aimed to verify the following hypotheses: (i) with the geographic substituted distribution of *Stipa* taxa, the rhizosphere soil AMF community structures and distribution and the content of glomalin will change; (ii) the AMF communities are largely influenced by climatic factors (mean annual precipitation [MAP] and mean annual temperature [MAT]) and plant

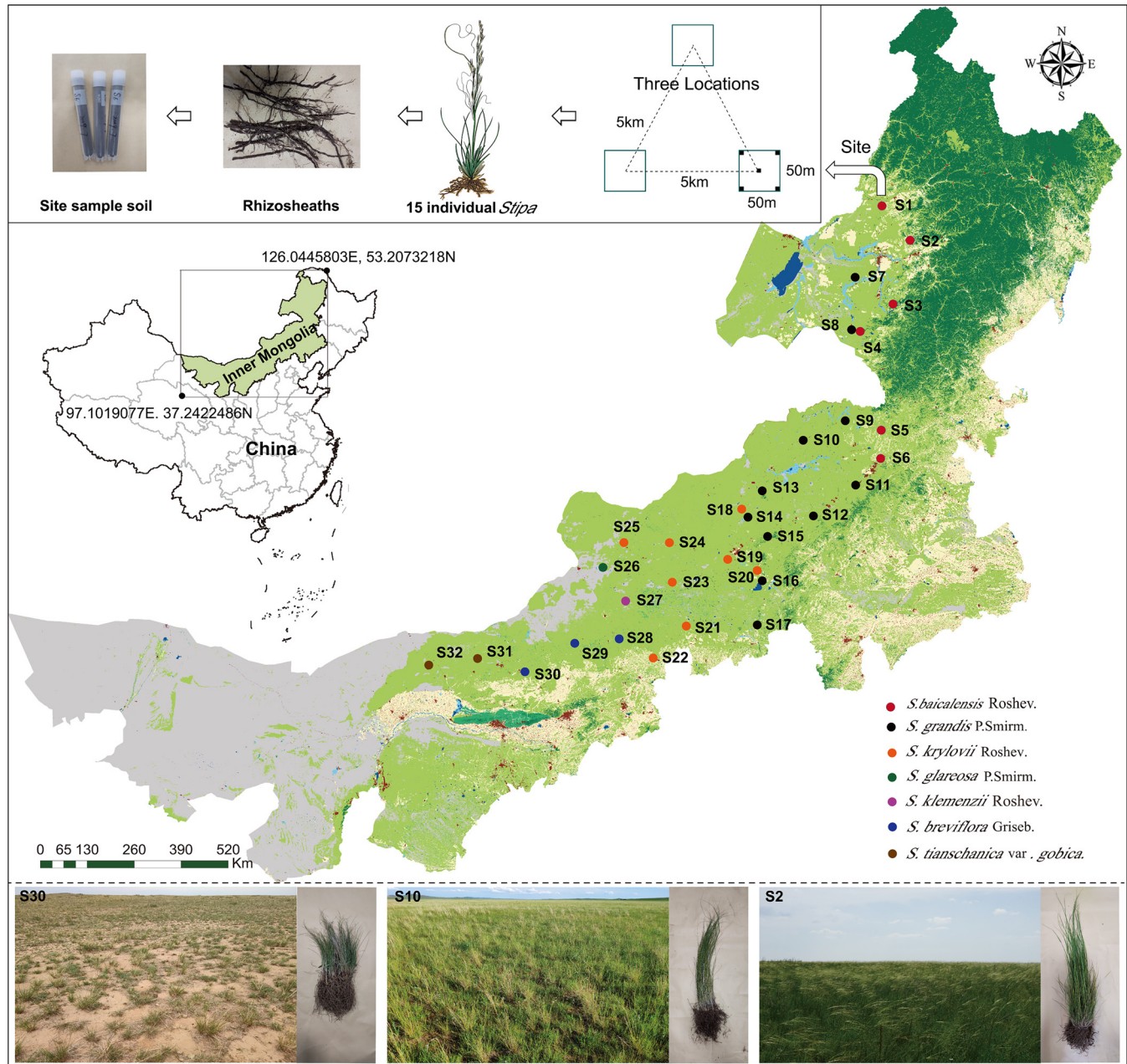

**FIG 1** Map of sample locations, landscapes, and schematic diagram of site sampling in arid and semiarid *Stipa* steppe. Each circle represents one sampling site, and each color represents a *Stipa* population. Samples collected from different sites and species are as follows: *Stipa baicalensis* Roshev (Sba1; sites S1 to S6), *Stipa grandis* P. Smirn (Sgr2; S7 to S17), *Stipa krylovii* Roshev (Skr3; S18 to S25), *Stipa glareosa* P. Smirn (Sgl4; S26), *Stipa klemenzii* Roshev (Skl5; S27), *Stipa breviflora* Griseb (Sbr6; S28 to S30), and *Stipa tianschanica* var. *gobica* (Stg7; S31 and S32). The map was obtained from BIGEMAP (www.bigemap.com).

community (*Stipa* taxa and biomass, above-ground diversity, and coverage); and (iii) the cooccurrence patterns of AMF taxa are significantly different among differing *Stipa* species.

## RESULTS

**Vegetation and soil characteristics of the study transect.** Different soil samples collected at different locations differed in physicochemical characteristics, vegetation indices, and climatic variables (Tables S1 and S2 in the supplemental material), and seven *Stipa* spp. were collected in this study. The MAP varied greatly among study sites, from 152.08 mm to 472.06 mm. Soils were mostly medium alkaline, with the average soil pH ranging from 6.06 to 8.74. The Shannon-Wiener index (PH), *Stipa* biomass (SB), and vegetation coverage (VC) of vegetation in the study area showed decreasing

**TABLE 1** Distribution of AMF genera and numbers of AMF species and OTUs among rhizosphere soils of seven *Stipa* species

| Genus | No. of: Species | OTUs in: Sba1 | Sgr2 | Skr3 | Sbr4 | Stg5 | Skl6 | Sgl7 | Total |
|---|---|---|---|---|---|---|---|---|---|
| *Acaulospora* | 1 | 0 | 2 | 0 | 0 | 0 | 0 | 0 | 2 |
| *Diversispora* | 4 | 12 | 14 | 7 | 0 | 0 | 2 | 0 | 15 |
| unclassified_f__Diversisporaceae | 1 | 2 | 3 | 2 | 0 | 1 | 0 | 0 | 4 |
| *Gigaspora* | 1 | 0 | 0 | 1 | 0 | 0 | 0 | 0 | 1 |
| *Scutellospora* | 3 | 5 | 9 | 4 | 0 | 0 | 0 | 0 | 9 |
| *Ambispora* | 1 | 2 | 2 | 1 | 0 | 0 | 0 | 0 | 2 |
| *Archaeospora* | 2 | 0 | 2 | 0 | 0 | 0 | 0 | 0 | 2 |
| unclassified_o__Archaeosporales | 1 | 0 | 3 | 0 | 0 | 0 | 0 | 0 | 3 |
| *Glomus* | 61 | 336 | 431 | 345 | 278 | 267 | 117 | 75 | 474 |
| *Paraglomus* | 7 | 12 | 65 | 31 | 4 | 0 | 0 | 0 | 73 |
| unclassified_c__Glomeromycetes | 1 | 34 | 634 | 139 | 12 | 10 | 3 | 2 | 674 |
| **Total:** | | | | | | | | | |
| 11 genera | 83 species | 403 | 1,165 | 530 | 294 | 278 | 122 | 77 | 1,259 OTUs |

trends from east to west, and they were significantly correlated with MAP and MAT. The organic phosphorus (OP), soil organic carbon (SOC), total nitrogen (TN), soil moisture (SM), VC, and SB rose in response to increasing MAP, while altitude, pH, and temperature showed decreasing trends. Pearson's correlation test explained significant relationships among *Stipa* taxa, glomalin, edaphic variables, and climatic and spatial factors (see Fig. 4). The easily extractable glomalin (EEG) and total glomalin (TG) decreased significantly with increasing altitude, pH, and MAT, whereas they increased with MAP, SOC, TN, OP, and total phosphorus (TP). In addition, glomalin was positively related to soil chemical elements.

**Variation and distribution among AMF communities in the rhizospheres of seven *Stipa* taxa.** For 96 samples, totals of 2,141,212 raw AMF 18SrDNA gene sequences and 2,101,443 high-quality reads were obtained by MiSeq sequencing (Table S3). The observed operational taxonomic units (OTUs) and rarefaction curves (Shannon, Chao1, and coverage indices) (Fig. S1) became saturated with an increasing number of samples, indicating adequate sampling intensity. Across all the samples, high-quality AMF sequences were grouped into 12,541 OTUs using the 97% sequence similarity cutoff, belonging to 83 AMF species, subordinate to 8 identified and 3 unidentified genera representing 9 families and 5 orders (Table S4).

The distribution and differentiation of AMF at the genus level and the relative abundances of AMF taxa among the rhizospheres in seven *Stipa* species are shown in Table 1 and Table S4, respectively. The OTU richness (1,259 OTUs) represented all 11 genera of the Glomeromycetes division. Among them, 2 each belonged to *Acaulospora*, *Ambispora*, and *Archaeospora*, 15 to *Diversispora*, 1 to *Gigaspora*, 9 to *Scutellospora*, 4 to unclassified_f__Diversisporaceae, 3 to unclassified_o__Archaeosporales, 674 to unclassified_c__Glomeromycetes, 73 to the *Paraglomus* genus, and 474 to the *Glomus* genus, accounting for a total percentage of 37.65% (Table 1). The highest abundance was found in the order Glomerales, with an average relative abundance of 90.19%, followed by Diversisporales (3.87%), unclassified_c__Glomeromycetes (3.61%), and Paraglomerales (1.95%) (Table S4). Of the 83 species, 61 belonged to the genus *Glomus*, accounting for a total percentage of 73.49%, and the relative abundance of *Glomus* remained relatively stable during the replacement distribution of *Stipa* from *S. baicalensis* (Sba1) to *S. krylovii* (Skr3) (Table 1 and Table S4). During the replacement distribution of Sba1 by *S. grandis* (Sgr2), the relative effect of the genus *Paraglomus* and family Paraglomeraceae (from 5.9% to 1.7%) decreased, while the genera *Acaulospora*, *Archaeospora*, and unclassified_o__Archaeosporales only occurred at the rhizosphere of Sgr2. When Skr3 replaced Sgr2, the relative abundances of the families Diversisporaceae (*Diversispora*), Paraglomeraceae (*Paraglomus*), and Gigasporaceae (*Scutellospora*) rapidly minimized to less than 1% with the occurrence of genera of *Gigaspora*. Finally, only Glomeraceae and Paraglomeraceae were distributed and functioning during the

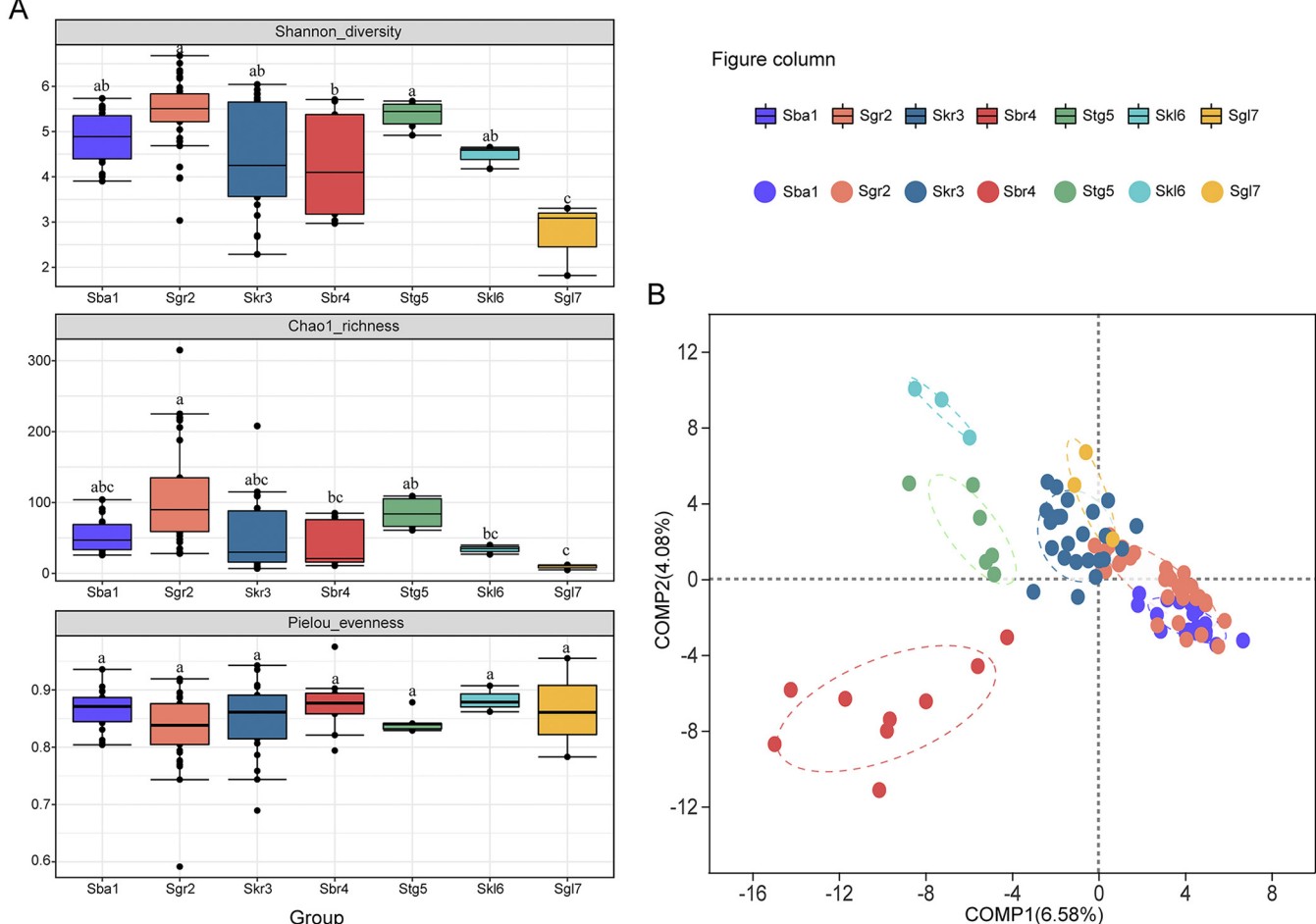

**FIG 2** (A) Variations and differentiation in the AMF community diversities in rhizosphere soils of seven *Stipa* species from different sites of arid and semiarid steppe. Different letters above the box-plot show significant difference among *Stipa* species based on one-way ANOVA ($P < 0.05$). (B) Partial least squares discriminant score plot of AMF communities among *Stipa* taxon groups.

replacement of Skr3 by the *Stipa* populations distributed on the desert steppe (*S. breviflora* [Sbr4], *S. gobica* [Stg5], *S. klemenzii* [Skl6], and *S. glareosa* [Sgl7]), while the family Diversisporaceae appeared at the Stg5 and Skl6 rhizospheres; the number of OTUs also decreased, from 530 to 294, 278, 112, and 77, respectively (Table 1). The relative roles of AMF taxa in the *Stipa* rhizosphere also varied at different sites of the same *Stipa* population, but there was no obvious turnover of AMF taxa (Fig. S2). Collectively, it was found that with the succession of the *Stipa* populations, the relative actions of some AMF taxa began to weaken at some sites in the same *Stipa* population, while the relative abundances of parts of the AMF taxa diminished further or even became nonfunctional.

**Changes of rhizosphere AMF diversity among different *Stipa* populations.** The Shannon-Weiner diversity index and Chao1 richness index for soil AMF OTUs of the seven *Stipa* taxa were significantly different ($P < 0.05$), but not the Pielou evenness index. The values for the Shannon-Wiener index ranged from 2.74 to 6.18, with a mean value of 4.82 (Table S3). The values for the richness (Chao index) and evenness (Pielou index) ranged from 9.33 to 181.67, with a mean value of 72.18, and from 0.74 to 0.91, with a mean of 0.85, respectively (Table S3). The AMF diversity and richness of the rhizosphere soils increased from Sba1 to Sgr2 and then gradually declined from Sgr2 to Sbr4, followed by a further increment when Sbr4 was replaced by Stg5 from east to west along the transect, and finally by decreases in the more arid areas (Skl6 and Sgl7) (Fig. 2A). The lowest Shannon-Wiener and Chao1 index values, which were notably distinct from those from

**TABLE 2** Differences in the amounts of glomalin and the ratios of glomalin fractions and soil organic carbon in different *Stipa* taxon rhizosphere soils

| Stipa taxon[a] | Mean value (g/kg or ratio) ± SD for[b]: | | | |
|---|---|---|---|---|
| | EEG | TG | EEG/SOC | TG/SOC |
| Sba1 | 4.081 ± 0.873 A | 7.396 ± 2.285 A | 0.208 ± 0.060 A | 0.378 ± 0.131 ABC |
| Sgr2 | 2.284 ± 0.928 B | 3.379 ± 1.049 B | 0.202 ± 0.072 A | 0.304 ± 0.095 BCD |
| Skr3 | 1.381 ± 0.900 BC | 2.737 ± 1.168 BC | 0.145 ± 0.108 AB | 0.286 ± 0.145 BCD |
| Sbr4 | 1.356 ± 0.408 BC | 2.717 ± 0.449 BC | 0.206 ± 0.078 A | 0.419 ± 0.138 AB |
| Stg5 | 1.302 ± 0.345 BC | 2.874 ± 0.182 B | 0.216 ± 0.088 A | 0.466 ± 0.091 A |
| Skl6 | 0.590 ± 0 CD | 1.448 ± 0 C | 0.107 ± 0 BC | 0.263 ± 0 CD |
| Sgl7 | 0.144 ± 0 D | 1.010 ± 0 C | 0.029 ± 0 C | 0.205 ± 0 D |
| F | 27.019 | 30.316 | 4.014 | 4.186 |
| P value | 0 | 0 | 0.001 | 0.001 |

[a]Sba1, *S. baicalensis*; Sgr2, *S. grandis*; Skr3, *S. krylovii*; Sgl7, *S. glareosa*; Skl6, *S. klemenzii*; Sbr4, *S. breviflora*; Stg5, *S. gobica*.
[b]Letters indicate significant differences evaluated by one-way ANOVA ($P < 0.05$). EEG, easily extractable glomalin; TG, total glomalin; SOC, soil organic carbon.

other *Stipa* taxa, appeared at Sgl7 ($P < 0.05$) (Fig. 2A). Descriptors of the AMF community were differentially correlated with ecological variables (Table S5). MAP, SB, and OP had significant positive effects on AMF diversity indices (Shannon and Chao1 indices), whereas MAT and PH remarkably and negatively influenced AMF diversity indices ($P < 0.05$). TN, EEG, and longitude only had a certain positive influence on the Shannon index. In addition, AM fungal Shannon diversity increased with the increase of most chemical nutrients. Moreover, positive relationships between Pielou indices and alkaline neutral available phosphorus (AP) and Ca were observed.

Permutational multivariate analysis of variance (PERMANOVA) showed that the AMF community composition was considerably influenced by the diverse *Stipa* taxa from which the soil samples were collected ($P < 0.001$), although the effect was small, with an $R^2$ of only 0.21 (Table S6). To assess the extent of the similarity of the AMF communities, partial least-squares discriminant analyses (PLS-DAs) were performed, and obvious separations between *Stipa* taxa were determined (Fig. 2B). The AMF communities in Sbr4, Stg5, and Skl6 clustered separately, indicating that the overall structures of the AMF communities in each group varied considerably. In addition, Sba1 was completely separated from Skr3, while Sgr2 was redundant with them as an intermediate transitional species.

**Soil glomalin analysis for *Stipa* rhizospheres.** Significant differences in the concentrations of the two glomalin fractions, EEG and TG, were observed in different *Stipa* rhizosphere soils (Table 2). The concentrations of both glomalin fractions decreased during the succession of Sbr4 (west) replacing Sba1 (east), with exceptionally high concentrations at Sba1. Typically, the concentrations of EEG and TG in the rhizospheres of *Stipa* species lay between $0.144 \text{ g} \cdot \text{kg}^{-1}$ and $4.081 \text{ g} \cdot \text{kg}^{-1}$ and between $1.010 \text{ g} \cdot \text{kg}^{-1}$ and $7.396 \text{ g} \cdot \text{kg}^{-1}$, respectively (Table S2). The TG/SOC and EEG/SOC ratios represent the soil carbon contribution rate, with mean values of 33.2% and 15.9%, respectively. The proportions of EEG/SOC (2.9% to 21.6%) and TG/SOC (20.5% to 46.6%) in the rhizospheres of *Stipa* species increased from east to west during the succession and distribution of *Stipa* species (Table S2). The lowest ratio of GRSP was found in the rhizosphere of Sgl7.

**AMF cooccurrence networks of *Stipa* rhizospheres.** To determine the complex AMF taxon associations in rhizosphere soils, networks were constructed for four of the *Stipa* taxa (Fig. 3). Multiple network topological metrics consistently showed that there were remarkable differences in AMF cooccurrence patterns among the four *Stipa* species. The average clustering coefficients (avgCC) for the Sba1, Sgr2, Skr3, and Sbr4 networks were 0.24, 0.27, 0.37, and 0.33, respectively. The average degrees (avgK) of Sba1, Sgr2, Skr3, and Sbr4 networks were 3.54, 1.02, 1.97, and 10.06, respectively. The average path distances (average geodesic distance, GD), of the Sba1, Sgr2, Skr3, and Sbr4 networks were 3.54, 3.88, 4.99, and 2.49, respectively. Compared to the networks of the other *Stipa* taxa, the Sgr2 and Skr3 networks had more positive connections (0.89/0.11 and 0.91/0.09, respectively), fewer nodes (60 and 86, respectively), and fewer links

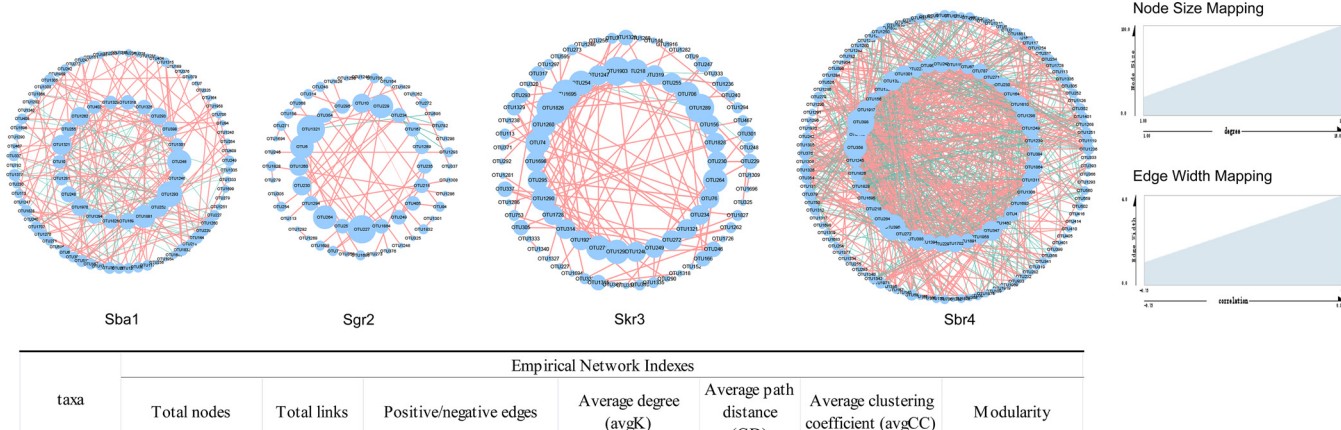

**FIG 3** Networks showing the different *Stipa* taxon cooccurrence patterns between AMF communities. Sba1, *S. baicalensis*; Sgr2, *S. grandis*; Skr3, *S. krylovii*; Sbr4, *S. breviflora*. SparCC correlation coefficients with a magnitude greater than 0.5 or less than −0.5 were considered. The size of each node is proportional to the number of connections (i.e., degree). The table below shows the corresponding network topology characteristics.

| taxa | Empirical Network Indexes | | | | | | |
|---|---|---|---|---|---|---|---|
| | Total nodes | Total links | Positive/negative edges | Average degree (avgK) | Average path distance (GD) | Average clustering coefficient (avgCC) | Modularity |
| Sba1 | 93 | 216 | 0.62/0.38 | 3.54 | 3.54 | 0.24 | 0.53 |
| Sgr2 | 60 | 66 | 0.89/0.11 | 1.02 | 3.88 | 0.27 | 0.75 |
| Skr3 | 86 | 117 | 0.91/0.09 | 1.97 | 4.99 | 0.37 | 0.8 |
| Sbr4 | 150 | 981 | 0.55/0.45 | 10.06 | 2.49 | 0.33 | 0.28 |

(66 and 117, respectively). Furthermore, the modularity values of the Sba1, Sgr2, Skr3, and Sbr4 networks were 0.53, 0.75, 0.8, and 0.28, respectively. This information indicated that Sbr4 possessed the most complex network and the lowest modularity, while Sgr2 and Skr3 had simple networks with high modularity. In summary, the Sgr2 and Skr3 networks were simple and connected with a high degree of modularity, while the Sbr4 network showed a pattern of high complexity and low modularity. All networks had more positive associations.

**Relationships between AMF community composition and ecological variables.** Four redundant variables (TP, Fe, Mg, and latitude) were eliminated based on variance inflation factor (VIF) analyses to inform the examination of the major environmental variables shaping AMF community structure and change by canonical correlations analysis (CCA) plots. The CCA plots (Fig. S3) revealed that the AMF community structure was primarily influenced by MAP ($r^2 = 0.6476$, $P < 0.01$), MAT ($r^2 = 0.7043$, $P < 0.01$), VC ($r^2 = 0.7026$, $P < 0.01$), and longitude ($r^2 = 0.5374$, $P < 0.01$) (Fig. S3). Meanwhile, Mantel test analyses declared that AMF community composition was highly correlated with MAP, EEG, VC, and Zn (Fig. 4). Finally, by combining the results of CCA plots with the Mantel test analyses, we determined that MAP and VC were the most dominant factors among other variables in the shifting AMF community structures. We found no strong effect of any ecological environmental factor on the *Glomus* genus composition (Fig. 4). The *Stipa* taxa and longitude had strong impacts on the *Diversispora* and *Scutellospora* genus compositions ($P < 0.01$). In addition, *Diversispora* was also significantly affected by MAT, whereas the *Paraglomus* genus composition was, to a larger extent, affected by certain ecological factors (e.g., MAP and so on).

## DISCUSSION

**Distribution and variations of the rhizosphere AMF communities along the *Stipa* population transition.** Our study demonstrated that the composition of *Stipa* rhizosphere soil AMF communities changed significantly with the change of succession gradients of *Stipa* populations in arid and semiarid grassland ecosystems. This may also rely on the climate and environmental gradients in the region. Climate (MAP and MAT) is one of the key elements influencing the AMF community structure, according to the Mantel test and CCA analyses of this study (Fig. 4 and Fig. S3). *Glomus* was the most prevalent and the most widely distributed genus across the seven *Stipa* species rhizospheres, contributing 37.6% of the total OTUs in the AMF community in this study. These results sustained previous reports

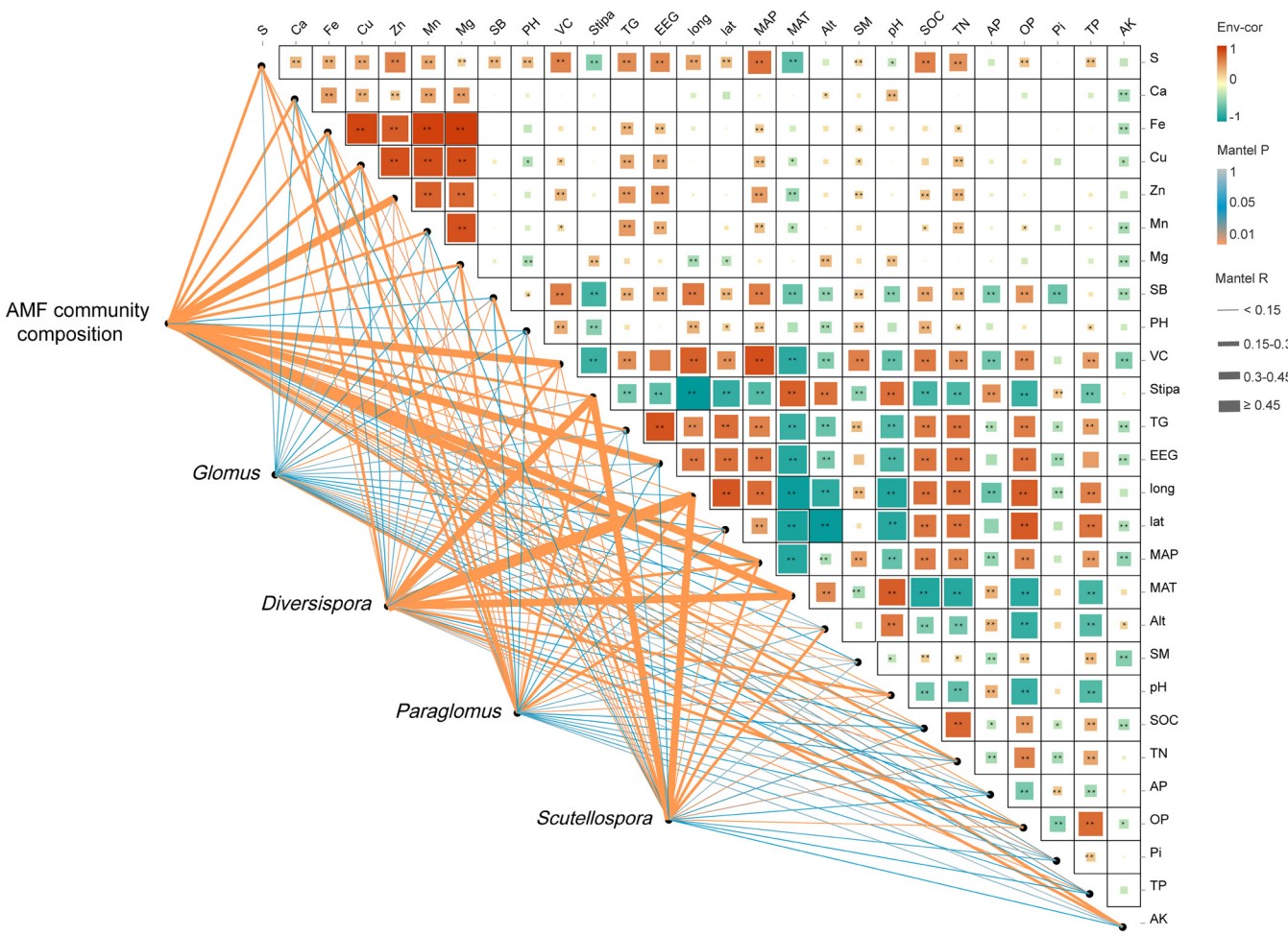

**FIG 4** Mantel correlations among AMF community compositions (OTU level), AMF genera, and environmental factors (climate, plant, and soil variables). Pearson correlation analysis was used among environmental factors. The width of an edge represents the size of the correlation coefficient (Mantel's *r*), while edge color represents the statistical significance based on Mantel's *P* value (*, $P < 0.05$ and **, $P < 0.01$). SOC, soil organic carbon; TN, total nitrogen; AP, alkaline neutral available phosphorus; AK, available potassium; Fe, Cu, Zn, Ca, S, and Mg, chemical elements; PH, Shannon-Wiener index of plot; VC, vegetation coverage; SB, biomass of *Stipa*; SM, soil moisture; OP, organic phosphorus; Pi, inorganic phosphorus; TP, total phosphorus; TG, total glomalin; EEG, easily extractable glomalin; long, longitude; lat, latitude, MAP and MAT, mean annual precipitation and temperature; Alt, altitude.

that the widely distributed genus *Glomus* had high abundance but low host specificity in arid environments (7). Members of the genus *Glomus*, for example, can adapt to a wide range of soil pH values but prefer to proliferate in alkaline and neutral soils, indicating that *Glomus* members may be owe to the broad ecological amplitude (41). Besides the genus *Glomus*, the genera *Paraglomus* and *Diversispora* were at the highest levels, with relative abundances of >1%; the genus *Paraglomus* decreased throughout the transition of the *Stipa* population from Sba1 to Skr3, while it rose at Sbr4, which disseminated farther westward (Table S4). Despite the similarities among dominant community compositions, some AMF groups in our research nevertheless displayed preferences in host-AMF associations in terms of families, genera, and OTUs. Phylotypes affiliated with the families Acaulosporaceae and Archaeosporaceae existed only in the rhizosphere of Sgr2, and the genus *Gigaspora* was only identified at Skr3. In addition, less than 0.1% of OTUs in AMF communities came from the families Ambisporaceae, Acaulosporaceae, and Gigasporaceae. There is evidence that those three families are overrepresented in the tropics rather than in northern temperate regions (42). Some species were replaced by others in drought environments; for example, the genera *Scutellospora*, *Ambispora*, *Archaeospora*, and *unclassified_o__Archaeosporales* did not emerge in the rhizospheres of *Stipa* in desert grassland (Sbr4, Sgl7, Skl6, and Stg5). It has been documented that these AMF species are also infrequent in arid regions (43), which indicates the scarcity of these taxa in highly arid areas where resources are usually limited.

The AMF compositions in rhizospheres across the seven *Stipa* taxa showed fair variance at the OTU level. These results suggested that the distribution patterns of AMF communities were heterogeneous and altered in response to changes in plant population at the OTU level and that some taxa may be identified as biomarkers for *Stipa* taxon rhizosphere soils.

**Diversity and network interaction patterns of soil AMF in rhizosphere soil of *Stipa* taxa.** We chose Shannon diversity, species richness, and Pielou's evenness as biodiversity metrics reflecting the dynamics of three aspects of community biodiversity. Considerable AMF diversity (Shannon index) and richness (Chao1 index) were identified in rhizosphere soils and with consistent change trends during the *Stipa* population turnover in space, while the Pielou index failed to discriminate significant changes. Namely, the index increased initially from Sba1 to Sgr2 and then progressively declined from Sgr2 to Sbr4, followed by a further increment after Sbr4 was replaced by Stg5 in the east-west direction of the transect and an eventual decrease in more arid areas (Sgl6 and Skl7) (Fig. 2A). With the least amount of precipitation and soil moisture, the Sgl7 rhizosphere AMF community's diversity and richness indices were lower than those of other *Stipa* species, revealing a high correlation between soil AMF diversity and climate, which was supported by the Pearson analyses between soil AMF diversity and MAP and MAT values (Table S5). Previous studies have revealed that water availability is a major constraint on biodiversity and ecosystem functioning in arid and semiarid ecosystems (14, 44), as a change in precipitation will directly influence the soil moisture, which affects the water utilization of vegetation and microorganisms (24, 45). Similar results have also been gathered showing that the diversity of AMF was highly dependent on precipitation in the agropastoral ecotone in arid and semiarid systems (46). From the viewpoint of the adaptability of AMF, an increase in temperature may be an environmental stressor (42). Our study found a significant negative correlation between AMF diversity and temperature, which is plausibly explained by the fact that a temperature rise will increase the availability of soil minerals and nutrients and decrease the amount of plant carbon that is allocated to mycorrhizal fungi, both of which have an impact on the AMF community (8, 42). In this study, the Shannon diversity of AMF demonstrated a significant negative correlation with soil pH. Most of the soils in the study sites were alkaline, and a high soil pH may inhibit the diversity of AMF to a certain extent by restricting the availability of soil nutrients (17). Previous studies on arid or semiarid grasslands have also attained similar results (34, 47). According to current research, soil nutrients were mostly responsible for the Shannon diversity shift of the AMF populations in the rhizosphere soils. For example, TN, OP, S, Fe, Cu, Zn, and Mn were found to be positively correlated with the Shannon index in rhizosphere soil (Table S5). Studies have demonstrated the significance of soil chemistry and macronutrients for microbial communities in arid and semiarid habitats (43, 48). Other studies have agreed that soil phosphorus has a significant role in determining the variety of AMF species in various grassland environments (49). Among the soil nutrients measured in our study, OP and AP showed positive correlations with the Shannon, Chao1, or Pielou index (Table S5). Considering the importance of phosphorus exchange in mycorrhizal associations, this was not unexpected (50). Apart from soil properties, vegetation biomass and diversity also cause changes in AMF diversity. Plant traits can transform substrate availability and the soil conditions, thereby altering the habitat of soil microbes (51). The aforementioned findings imply that the AMF diversity in the rhizosphere soil of *Stipa* across a large spatial scale was the combined outcome of multiple complicated ecological processes.

**OTU cooccurrences.** OTU cooccurrences can affirm the interactions among species, which is crucial for understanding the diversity, structure, and function of AMF in rhizosphere soil (52, 53). The numbers of nodes and edges and the average degree of a biological community cooccurrence network report the complexity of community connections (51, 53). Comparison among the AMF communities of four *Stipa* taxon rhizosphere soils revealed that the numbers of nodes and edges and the average degree of the Sbr4 network were the highest, suggesting that the most complex relationships were among the AMF communities of Sbr4. Through the comparison of these

topological characteristics, it was found that the networks of Sgr2 and Skr3 were simpler, especially the Sgr2 network, which had lower network connectivity. In the present study, AMF formed positive cooccurrence networks in the rhizospheres of all *Stipa* species, which indicated that the pattern promoted interactions. These positive interactions indicated a positive synergistic response of root zone-related AMF to harsh environments (54). It is worth noting that the proportions of positive correlations between the Sgr2 and Skr3 networks were extremely high and that communities in which the majority of members are linked by positive associations are considered to be unstable because they are prone to a common response in the face of environmental changes, while moderate negative associations contribute to community stability (55). A network composed of shorter paths possesses "small-world" properties associated with a faster response to disturbances (56, 57). The Sbr4 network not only had the shortest average path but also had the smallest modularity, indicating that when the external environment is disturbed, the AMF network can transmit the environmental disturbance signal to the whole network in a very short time, so the network can respond faster to environmental changes. In addition, Sbr4's network was the most complex, and its negative connections might stabilize the coosillation in the community and promote the stability of the network (55). Therefore, it was considered that the network was relatively stable, which also supported the central ecological belief that complexity begets stability (57). Future studies should pay more attention to the AMF species resources in the rhizosphere of vegetation in desert steppes.

**Distribution pattern of glomalin-related soil protein along the *Stipa* population transition.** As a metabolite of AMF, GRSP is of exceptional importance in contributing to soil C sequestration in arid and semiarid ecosystems (14, 17). In this study, the GRSP contents and EEG/SOC and TG/SOC ratios in the rhizospheres of *Stipa* taxa revealed high variation and spatial dynamics. The concentrations of GRSP gradually decreased, while in contrast, the contribution to soil carbon (TG/SOC and EEG/SOC ratios) gradually increased during the process of Stg5 (west) replacing Sba1 (east). Interestingly, Sgl7 and Skl6, which remained the lowest in GRSP content, had the lowest MAP. Considering that the GRSP content showed a strong positive correlation with organic phosphorus, TN, SOC, and pH but a negative correlation with MAT and pH and that it decreased with the declination of MAP (Fig. 4), we believe that soil nutrients might not be fully utilized due to low MAP and high pH. Therefore, the GRSP contents in *Stipa* rhizospheres were lower in more arid and alkaline areas. Precipitation will have a considerable influence on ecosystem production, since water is the primary resource that limits the metabolism of grassland ecosystems (43). Consequently, the content of glomalin and its ratio to soil carbon in arid and semiarid grasslands may be influenced to a larger extent by precipitation. Furthermore, when assessed by the ratio of soil carbon, the contribution of glomalin to soil carbon storage was higher in the arid areas (17).

**Responses of AMF communities and populations to ecological variables.** The changes of AMF community characteristics in different *Stipa* species rhizosphere soils are complex and influenced by multiple abiotic and biotic factors. In arid and semiarid grassland ecosystems, soil pH and climatic elements have been proposed as important influences, since they are closely related to plant species and AMF community characteristics (33, 43, 45, 49). In our study, we comprehensively considered the impacts of climate, geography, vegetation, and physicochemical parameters on AMF communities and populations. Among the climatic factors, temperature, humidity, and their combination substantially affect the zonal geographic distribution of terrestrial biological communities (20), and hydrothermal conditions also exert a key impact on the growth of AMF and host plants (24). Our study showed that MAP and MAT influenced the variation of AMF communities in the rhizospheres of *Stipa* across the region remarkably (Fig. S3). Since soil moisture plays a decisive role in fungal metabolism and AMF is sensitive to the abiotic environment, a change of precipitation that leads to a variation in soil moisture will in turn direct AMF to groups that favor plant growth under drought conditions (30, 33, 58). The present study by Gao et al. evaluated the effects of the conditions of climate warming and increased precipitation on AMF communities in the

semiarid grassland of northern China (24). It was found that AMF communities were notably affected by the increase in precipitation, but not by the temperature rise. This is distinct from the natural environment survey. Moreover, higher correlations were found between biotic factors and AMF communities in the rhizosphere soils of *Stipa* species. AMF community composition was affected by vegetation coverage (Fig. 4), indicating that water availability affects aboveground productivity and further influences AMF. In this study, different AMF taxa (genera) responded differently to eco-environmental factors, which may in turn illustrate the spatial disparity of AMF diversity and community composition. Certain studies have revealed that geographical distance (longitude and latitude) may also affect the community structure of AMF (59). In this study, we found that longitude exerted strong effects on *Diversispora* and *Scutellospora* genus composition. Considering that AMF form the association between the root system of host plants and the soil environment, their taxa are believed to be directly affected by host plants. Both the *Diversispora* and *Scutellospora* had high correlations with the *Stipa* taxa in this study (Fig. 4). The PERMANOVA and PLS-DA analyses found (Fig. 2B and Table S6) that the AMF communities had a separation phenomenon and their compositions were significantly affected by the *Stipa* taxa, but the impact index was small, which could be rationally attributed to the fact that all the species are members of the same family, *Poaceae* (7). Other studies in semiarid areas hypothesized that various host plants could affect the presence of some AMF species, which preferentially react to vegetation change at the species level (26, 47). This indicates that plant ecological groups presumably affect the relative abundance rather than the absence or existence of AMF taxa (47). In addition to being strongly tied to plants and AMF species, host plants' effects on AMF community structure were also influenced by a variety of habitat factors in our study, including soil characteristics like Zn and EEG, climatic factors like MAT and MAP, and longitude (Fig. 4). As ecological factors affecting AMF in nature, host plants have complex interactions with other factors (Fig. 4). These results suggest that the influence of plant species on the community composition of rhizosphere AMF is not as strong as that of the abiotic factors within a natural ecosystem and that the role of the host will eventually be weakened, which means that the rhizosphere AMF community shows low host specificity but fair heterogeneity in the distribution process.

In summary, this study for the first time depicted the variations, species distribution, and influencing factors of rhizosphere soil AMF communities along the alternative distribution of *Stipa* populations on an east-west transect across arid and semiarid grassland systems. The results indicated that the GRSP, species interaction, species, and quantity of AMF in rhizospheres changed with the substitution distribution of the *Stipa* populations, although there was no clear substitution of AMF taxa, including the same *Stipa* population at different sites. Additionally, the heterogeneous distribution and community structures of rhizospheres were along combined gradients of plant population and environmental variables across the arid and semiarid region of northern China, and various AMF groups (genera) responded diversely to eco-environmental factors. Moreover, network analysis indicated that the cooccurrence patterns of AMF taxa in the rhizospheres of different *Stipa* species were significantly different. Our work closes a knowledge gap in the regional scales of arid and semiarid grasslands' AMF community diversity and composition in a variety of *Stipa* species rhizospheres, which might serve as a useful guide for local ecological environment conservation.

## MATERIALS AND METHODS

**Study transect description.** In this study, an Inner Mongolia grassland (41°52′ to 50°12′N, 108°46′ to 120°21′E) in northern China was selected as the study area. The region is characterized by a continental arid and semiarid monsoonal climate accompanied by abrupt changes of temperature in the middle temperate zone (40, 60). The climate is distributed in a belt and gradually transits from a semihumid to semiarid and arid areas from east to west (5). The area is characterized by prevailing winds from multiple directions, with a northwest wind prevailing in winter and southerly and southeast winds in summer (61). *Stipa* spp. are the dominant species in the study steppe, and *Leymus chinensis* and *Filifolium sibiricum* are also commonly developed (49, 61).

**Collection of plant and rhizosphere soil samples.** Seven zonal *Stipa* species distributed over 32 sample sites with nearly natural plant communities or only a few grazing signs were selected along a 1,700-km transect in Inner Mongolia (Fig. 1). The specific site information is displayed in Table S1: the population of *S. baicalensis* (Sba1, sites S1 to S6) was distributed in meadow steppe, while *S. grandis* (Sgr2, S7 to S17) and *S. krylovii* (Skr3, S18 to S25) populations were predominantly distributed in typical steppe. Some Sgr2 populations (S11 and S12) infiltrated into meadow steppe, and part of the Skr3 population (S25) infiltrated into desert steppe. The *S. breviflora* (Sbr4, S28 to S30), *S. gobica* (Stg5, S31 to S32), *S. glareosa* (Sgl7, S26) and *S. klemenzii* (Skl6, S27) populations were all distributed in desert steppe. According to the growth rhythms of the seven *Stipa* populations, samples were collected from west to east along the horizontal sample belt from early August 2019 to ensure that *Stipa* populations were in uniformly mature phenological periods. During the sampling process, *Stipa* species were identified according to the third edition of *Flora Intramongolica* (62). At each sampling site, three large rectangular locations (50 by 50 m) with intervals of more than 5 km were established, and five sampling plots (1 m by 1 m) were set within each location, one at each corner and one in the center (Fig. 1). A vegetation survey, including plant diversity, number, vegetation coverage, and biomass of *Stipa*, as well as a calculation of the Shannon-Wiener index ($H' = -\Sigma_{i=1}^{r}[Pi\mathrm{Ln}Pi]$, where Pi represents the ratio of the number of each species to the total number of all species, and r is the number of species observed in the quadrat) was conducted in each plot. Five individual *Stipa* plants with similar aboveground biomass were selected from each location to reduce the influence of age on the diversity and composition of soil AMF community. A total of 15 individual *Stipa* plants with intact aboveground structures and belowground root systems were excavated at each site. The extremely loose soil on the 15 individual *Stipa* roots from each respective site was removed, and the rhizosphere soil (63) adhered to the rhizosheath was collected with a disposable brush and then divided into three equal parts. In all, 32 × 3 rhizosphere soil samples were collected. One part was packed into three cryopreservation tubes, immediately transferred to a liquid nitrogen tank, and stored at −80°C before being returned to the laboratory for 18S rRNA high-throughput-sequencing analysis. Each fresh soil sample was divided into two portions. One portion was sieved to <2 mm and stored at 4°C prior to analysis of physiochemical properties and GRSP. The second portion was stored in a freezer at −20°C after being put through a 200-mesh sieve (pore size of 0.074 mm) prior to analysis of soil chemical elements.

**Ecological environment factor analysis.** The pH value was measured with a soil pH detector (SKW 500 kit) from Palintest Ltd. After digestion with $H_2SO_4$ ($\rho = 1.8419$ g · $L^{-1}$), the soil samples were assayed to determine the total nitrogen (TN) using an automatic Kjeldahl apparatus (K1100 Hanon; Haineng Future Technology Group Co., Ltd., China). The soil organic carbon (SOC) was determined by dichromate oxidation (64). Sodium bicarbonate (0.05 mol · $L^{-1}$) was used for extraction, and the molybdenum-antimony anticolorimetric method was employed to determine the Olsen phosphorus (AP). Available potassium (AK), total phosphorus (TP), organic phosphorus (OP), and inorganic phosphorus (Pi) were measured with a kit from Suzhou Keming Biotechnology Co., Ltd., Suzhou, China, following the manufacturer's instructions. The soil moisture content was detected rapidly by the WET kit (Delta-T Devices Ltd., UK). Soil samples were digested by using the MARS 6 system (CEM Co., Ltd., USA), strictly following the experimental manual (HJ 832-2017; Ministry of Environmental Protection [MEP], China). Determination of soil chemical elements (S, Zn, Ca, Cu, Mg, Fe, Na, and Mn) relied on inductively coupled plasma optical emission spectrometry (ICP-OES) using the PlasmaQuant PQ9000 emission spectrometer (Jena Analytical Instrument Co., Ltd., Germany) after filtration through a filtering membrane (0.45 $\mu$m). The altitude, longitude, and latitude of each site were registered using GPS. The mean annual temperature (MAT) and mean annual precipitation (MAP) were gathered from NASA's GES DISC (https://disc.gsfc.nasa.gov/).

**Soil glomalin analysis.** Easily extractable glomalin (EEG) was extracted from 0.15 g soil with 20 mmol · $L^{-1}$ sodium citrate, pH 7.0, at 103 kPa and 121°C for 30 min, and total glomalin (TG) was extracted with 50 mmol · $L^{-1}$ sodium citrate, pH 8.0, at 103 kPa and 121°C for 60 min, using an autoclave. The clear supernatant liquid was collected after centrifugation, and the procedure was repeated until the extract was straw colored. An Amicon Ultra filter was used to remove impurities. EEG and TG were quantified by the bicinchoninic acid (BCA) assay. Results are expressed in milligrams of GRSP per gram of soil.

**DNA extraction, PCR amplification, and Illumina sequencing of AMF.** Following the manufacturer's instructions, triplicate genomic DNA samples were extracted from 0.5-g composite soil samples using the E.Z.N.A. soil DNA kit (Omega Bio-tek, Norcross, GA, USA). The universal primers AWV4-5NF (5′-AAGCTCGTAGTTGAATTTCG-3′) and AMDGR (5′-CCCAACTATCCCTATTAATCAT-3′) (24) were introduced using an ABI GeneAmp 9700 PCR thermocycler (ABI, USA). The DNA concentration and purity were detected with the NanoDrop 2000 (Thermo Scientific, Wilmington, DE, USA), and the DNA extraction quality was detected by 1% agarose gel electrophoresis (5 V/cm for 20 min). The PCRs were performed in triplicate, and negative controls were included during each cycle of PCR. Thermocycling was begun by an initial denaturation at 95°C for 3 min, followed by 30 cycles of denaturing at 95°C for 30 s, annealing at 55°C for 30 s, and extension at 72°C for 45 s, a single extension at 72°C for 10 min, and termination at 10°C. The PCR products were purified using the AxyPrep DNA gel extraction kit (Axygen Biosciences, Union City, CA, USA) before they were quantified using the Quantus fluorometer (Promega, USA). The amplified PCR products were purified using the Nextflex rapid DNA-Seq kit to form a sequencing library, which was sequenced using the Illumina MiSeq PE300 platform (Illumina, San Diego, CA, USA).

**Bioinformatics analysis.** We demultiplexed and quality filtered the raw sequences from the MiSeq platform using Trimmomatic (http://www.usadellab.org/) (65) and merged them with FLASH 1.2.11. Subsequently, the UPARSE (version 7.0.1090) algorithm was applied; high-quality sequences were clustered into operational taxonomic units (OTUs) with a 97% similarity cutoff, and chimeric sequences were identified and removed. A representative sequence was aligned, using the RDP Classifier (http://rdp.cme

.msu.edu/) in the QIIME platform, against the AMF 18S rRNA database (Maarj*AM* 081/AM; https://www .maarjam.ut.ee) at a confidence threshold of 0.7 to facilitate the comparison between AMF sequences in this study and those of other studies. To reduce biases caused by different sequencing depths, all samples were rarefied to the lowest sequencing depth possible (9,953 sequences) and evaluated using MOTHUR (version 1.30.1). The raw sequencing data were filtered and trimmed, and denoised reads merged by the mergePairs DADA2 plugin, using QIIME2 to correct for overinflated alpha diversity. Then, alpha analyses were performed to characterize AMF richness, diversity, and evenness (66, 67).

**Statistical analysis.** One-way analysis of variance (ANOVA) in IBM SPSS Statistics 26.0 was exploited to assess the significance, and a $P$ value of $<0.05$ was set as the threshold of significance. Partial least squares discriminant analysis (PLS-DA) was also performed to compare AMF compositions among *Stipa* taxa. To determine whether the *Stipa* species directly and/or indirectly affected the AMF community composition, permutational multivariate analysis of variance (PERMANOVA, $P < 0.05$) was performed with the Adonis (999 permutations) function by using the vegan package (68). To test the relative significance of ecological environmental factors in AMF community composition variation in the rhizospheres, the Pearson correlation and Mantel test were performed by using the vegan package implemented in R (69). In canonical correlations analysis (CCA) modeling, the variance inflation factor (VIF) was used to remove redundant variables from explanatory variables. Meanwhile, only variables that were determined to be significant by the Mantel test ($P < 0.05$) were included in CCA analyses. To simplify the networks for more effective visualization and unified analysis criteria, OTUs with a frequency lower than 1/3 were removed, and Stg5, Skl6, and Sgl7 rhizosphere soils were excluded from the analysis due to the limited number of samples. We built the cooccurrence network of AM fungi using the Python-based SparCC (https://github.com/shafferm/fast_sparCC) correlation method (70). Next, SparCC was run with 1,000 bootstraps and its $P$ value was calculated by false discovery rate correction (71). Only significant ($P < 0.05$) and strongly positive (SparCC values greater than 0.5 and less than $-0.5$) correlations were considered. Finally, Cytoscape version 3.8.2 was used for network visualization.

**Data availability.** The raw sequencing data for AMF 18S rRNA reads of each rhizosphere soil sample have been deposited at NCBI under accession numbers SRR18034159 and SRR18034254.

## SUPPLEMENTAL MATERIAL

Supplemental material is available online only.

**SUPPLEMENTAL FILE 1**, PDF file, 1.7 MB.

## ACKNOWLEDGMENTS

This work was supported by the National Natural Science Foundation of China (grant number 31760005), the Science and Technology Plan of Inner Mongolia Autonomous Region (grant number 2020GG0079) and the Natural Science Foundation of Inner Mongolia Autonomous Region of China (2021BS03030).

All authors made corresponding contributions. Xiaodan Ma wrote the manuscript with significant assistance and comments from Yuying Bao. We thank all of the people who helped during the field and laboratory work of this study. We thank the anonymous reviewers for their help in improving the manuscript with constructive comments.

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
