## [Reviewer comments · Microbiology Spectrum]

Microbiology Spectrum

Changes of AM Fungi Community and Glomalin in the Rhizosphere Along the Distribution Gradients of Zonal Stipa Population Across the Arid and Semiarid Steppe.

Xiaodan Ma, Jingpeng Li, Fucheng Ding, Yaxin Zheng, Lumeng Chao, Haijing Liu, Haijing Liu, Hanting Qu, and Yuying Bao

Corresponding Author(s): Xiaodan Ma, Inner Mongolia University

Review Timeline:

Submission Date:	April 24, 2022
Editorial Decision:	August 17, 2022
Revision Received:	September 13, 2022
Accepted:	September 20, 2022

Editor: Christina Cuomo

Reviewer(s): Disclosure of reviewer identity is with reference to reviewer comments included in decision letter(s). The following individuals involved in review of your submission have agreed to reveal their identity: Bo Maxwell Stevens (Reviewer #1); Hao Tan (Reviewer #2)

Transaction Report:

DOI: <https://doi.org/10.1128/spectrum.01489-22>

August 17, 2022

Dr. Xiaodan Ma
Inner Mongolia University
School of Life Sciences, Inner Mongolia University
Hohhot
China

Re: Spectrum01489-22 (**Changes of AM Fungi Community and Glomalin in the Root-zone Along the Distribution Gradients of Zonal *Stipa* Population Across the Arid and Semiarid Steppe.**)

Dear Dr. Xiaodan Ma:

Thank you for submitting your manuscript to Microbiology Spectrum. Two reviewers have provided feedback that I would like you to address in a revision. In particular, the comments of reviewer 1 (see attached review) about the soil sampling with regards to controlling for biotic factors are critical to address.

Link Not Available

Sincerely,

Christina Cuomo

Journals Department
Reviewer comments:

Reviewer #1 (Comments for the Author):

The bioinformatics could also be improved. I highly recommend using DADA2 for the bioinformatics analysis, it is the current gold standard for Illumina MiSeq data because it helps correct for over-inflated alpha diversity (Straub et al. 2020). QIIME2 makes this easy (Bolyen et al. 2018). A new taxonomic classifier might help as well (Curry et al. 2022).

Abstract

Line 17: Might what to reword as "Exactly how AMF ..." - sounds better reading it

Line 19: What exactly is meant by "adaptive relationship between plants"

Line 20: I'm not exactly sure what is meant by "zonal"

Line 21: "abundant resources (83 species) of AMF were isolated" -- not accurate wording

Line 23: "quantity" - do you mean relative abundance?

Introduction

Line 36: Use "account"

Line 37: Should be "Earth"

Line 37: I would avoid using "aroused", maybe use "attracted" instead

Methods

Line 133: Do you mean "habitats"?

Line 209: Specify the sampling depth

Figures

Figure 2: I don't think you need to Have the family and the table. I typically just report the genus for AMF, you could combine this with a figure of the alpha diversity or beta diversity ordination.

Figure 3: I don't think this figure is necessary.

Figure 4: Use the full name (e.g. Faith's PD) and define the acronym somewhere in the methods. It might be useful to report PERMANOVA results in this figure. I don't think you need the NMDS figure, just the Partial least square discriminant plot. Also, I don't think there is a need for all three alpha diversity metrics, maybe just pick the most interesting and put the rest in the supplemental information (describe all in the methods and results section, though). Also, on top of the boxplots, it may be easier to use letters from a Tukey's Honest Significant test. It may be helpful to number the plant species (with a key Table 1) instead of using abbreviations; the abbreviations are so similar they blend together.

References

Bolyen, Evan, Jai Ram Rideout, Matthew R. Dillon, Nicholas A. Bokulich, Christian Abnet, Gabriel A. Al-Ghalith, Harriet Alexander, et al. 2018. "QIIME 2: Reproducible, Interactive, Scalable, and Extensible Microbiome Data Science." e27295v2. PeerJ Inc. <https://peerj.com/preprints/27295>.

Curry, Kristen D., Qi Wang, Michael G. Nute, Alona Tyshaieva, Elizabeth Reeves, Sirena Soriano, Qinglong Wu, et al. 2022. "Emu: Species-Level Microbial Community Profiling of Full-Length 16S rRNA Oxford Nanopore Sequencing Data." *Nature Methods* 19 (7): 845-53.

Straub, Daniel, Nia Blackwell, Adrian Langarica-Fuentes, Alexander Peltzer, Sven Nahnsen, and Sara Kleindienst. 2020. "Interpretations of Environmental Microbial Community Studies Are Biased by the Selected 16S rRNA (Gene) Amplicon Sequencing Pipeline." *Frontiers in Microbiology* 0. <https://doi.org/10.3389/fmicb.2020.550420>.

Reviewer #2 (Comments for the Author):

This manuscript reported the associations between AMF and *Stipa* distribution in combination with the influences on glomalin and glomalin-related soil proteins. This study was solidly supported by a large amount of samples and data from multiple aspects. The results are very informative and complete. The topic and results are of high novelty. The writing and language of the manuscript as well as some of the bioinformatic analyses need improvement.

1. Structure of manuscript:

The entire manuscript has not been formatted into the requested layout of *Microbiology Spectrum*. Should follow the styles of ASM journals: Introduction, Results, Discussion, than last Materials and Methods. The English usage such as "inclined to decrease" is wrong. Please use a professional company or native speaker to polish the English of this manuscript.

2. Introduction

(1) I feel the introduction too long. One and a half page is enough. Please be concise.

(2) Why *Stipa* plants are important to be involved in this study? What's the ecological importance of the plants themselves, not only the AMF. This need to state in the introduction. Similar for the glomalin. Why is it important to be included into this study?

3. The methods part:

(1) "The region is characterized by continental arid and semi-arid monsoonal climate accompanied by abrupt changes of temperature in the middle temperate zone." "The climate is distributed in a belt, and gradually transits from semi humid to 1 semi-arid and arid area from east to west." "The area is characterized by multi prevailing winds, with northwest wind prevailing in winter and southerly and southeast wind in summer." " *Stipa* spp. is the dominant species in the study steppe, and *Leymus chinensis* and *Filifolium sibiricum* are also commonly developed." The four sentences need to provide literature references to support these claims.

(2) Microbial community diversity and richness have been determined, however community evenness is missing. It is important to measure community evenness, so that we can see if the change of diversity is majorly contributed by taxonomic richness or community evenness. I recommend to use Pielou's Evenness (for example, doi: 10.1007/s00253-022-12038-2; 10.1128/Spectrum.00229-21).

(3) Why use CCA plots not RDA plots? I find the value of the CCA1 axis and CCA2 axis are both very low (<5%), while other studies always have CCA axes >20%. The author can try RDA and see if they can get a better result.

(4) For co-occurrence network of microbial community, SparCC model is better than the models based on simpler algorithms such as Pearson, Spearman, and Kendall. I recommend to adopt SparCC to re-calculate the network as the method used in doi 10.1016/j.biortech.2022.127549, in order to make the results more reliable.

(5) The authors said "AMF-specific primers were designed based on the most variable segment of the small subunit (SSU) rRNA gene region." The authors mean that they designed the primers by themselves?

Staff Comments:

Preparing Revision Guidelines

Please return the manuscript within 60 days; if you cannot complete the modification within this time period, please contact me. If you do not wish to modify the manuscript and prefer to submit it to another journal, please notify me of your decision immediately so that the manuscript may be formally withdrawn from consideration by Microbiology Spectrum.

Overall comments

The authors report the soil arbuscular mycorrhizal fungal (AMF) microbiome associated with varieties of a *Stipa* throughout Inner Mongolia. The molecular methods are worthwhile and the study as a whole could be interesting if presented within a different ecological framework. However, unfortunately, I cannot recommend this for publication in this Journal. It's not clear to me if these samples are bulk soil or rhizosphere soil. Sometimes AMF in the soil respond to abiotic factors, rather than biotic factors. With the way this study was performed, it would be difficult to separate these factors because the plants are not evenly distributed throughout the study area. The manuscript could benefit from improved writing and clarification of the results. There exists many confusing and atypical terminology throughout the manuscript.

The bioinformatics could also be improved. I highly recommend using DADA2 for the bioinformatics analysis, it is the current gold standard for Illumina MiSeq data because it helps correct for over-inflated alpha diversity (Straub et al. 2020). QIIME2 makes this easy (Bolyen et al. 2018). A new taxonomic classifier might help as well (Curry et al. 2022).

Minor comments

Abstract

Line 17: Might what to reword as "Exactly how AMF ..." — sounds better reading it

Line 19: What exactly is meant by "adaptive relationship between plants"

Line 20: I'm not exactly sure what is meant by "zonal"

Line 21: "abundant resources (83 species) of AMF were isolated" — not accurate wording

Line 23: "quantity" — do you mean relative abundance?

Introduction

Line 36: Use "account"

Line 37: Should be "Earth"

Line 37: I would avoid using "aroused", maybe use "attracted" instead

Methods

Line 133: Do you mean "habitats"?

Line 209: Specify the sampling depth

Figures

Figure 2: I don't think you need to Have the family and the table. I typically just report the genus for AMF, you could combine this with a figure of the alpha diversity or beta diversity ordination.

Figure 3: I don't think this figure is necessary.

Figure 4: Use the full name (e.g. Faith's PD) and define the acronym somewhere in the methods. It might be useful to report PERMANOVA results in this figure. I don't think you need the NMDS figure, just the Partial least square discriminant plot. Also, I don't think there is a need for all three alpha diversity metrics, maybe just pick the most interesting and put the rest in the supplemental information (describe all in the methods and results section, though). Also, on top of the boxplots, it may be easier to use letters from a Tukey's Honest Significant test. It may be helpful to number the plant species (with a key Table 1) instead of using abbreviations; the abbreviations are so similar they blend together.

References

- Bolyen, Evan, Jai Ram Rideout, Matthew R. Dillon, Nicholas A. Bokulich, Christian Abnet, Gabriel A. Al-Ghalith, Harriet Alexander, et al. 2018. "QIIME 2: Reproducible, Interactive, Scalable, and Extensible Microbiome Data Science." e27295v2. PeerJ Inc. <https://peerj.com/preprints/27295>.
- Curry, Kristen D., Qi Wang, Michael G. Nute, Alona Tyshaieva, Elizabeth Reeves, Sirena Soriano, Qinglong Wu, et al. 2022. "Emu: Species-Level Microbial Community Profiling of Full-Length 16S rRNA Oxford Nanopore Sequencing Data." *Nature Methods* 19 (7): 845–53.
- Straub, Daniel, Nia Blackwell, Adrian Langarica-Fuentes, Alexander Peltzer, Sven Nahnsen, and Sara Kleindienst. 2020. "Interpretations of Environmental Microbial Community Studies Are Biased by the Selected 16S rRNA (Gene) Amplicon Sequencing Pipeline." *Frontiers in Microbiology* 0. <https://doi.org/10.3389/fmicb.2020.550420>.

Dear Reviewer:

Thank you for your letter and comments concerning our manuscript entitled “Changes of AM Fungi Community and Glomalin in the Rhizosphere Along the Distribution Gradients of Zonal *Stipa* Population Across the Arid and Semiarid Steppe” (Spectrum01489-22). I have received new suggestions and those comments are very detailed, professional and very helpful for revising and improving our paper, as well as the important guiding significance to our researches. We strongly cherish this opportunity that you gave us to further revise. Based on your comments and requests, we have studied comments carefully and have made correction which we hope meet with approval.

For editors and reviewers convenience, we have highlighted the changes in blue according to reviewer comments in the annotated version of the revised manuscript (Revision, changes marked). In addition, the manuscript has been polished in English, but there is no specific mark. I highlight the questions and comments in red. The main corrections in the paper and the responds to the reviewer’s comments are as flowing:

Reviewer #1:

1. Questions and comments: It’s not clear to me if these samples are bulk soil or rhizosphere soil.

Response 1: After shaking off the loose soil, we use a disposable brush to brush the soil still adhered to the rhizosheath until no more soil was brushed down, and collect the root-zone soil samples. We do not use PBS to wash root segment and collect soil by centrifugation. It may be more suitable to call it root-zone soil. We mainly refer to the definitions in the following articles (Edwards et al. 2015, Shi et al. 2019). Although *Stipa* has rhizosheath, the surrounding soil is not as loose as desert soil. In many related scientific studies (Gremion, Chatzinotas and Harms. 2003; Lavecchia et al. 2015),

loosing (shaking off) soil around easily removed roots is called ‘Bulk Soil’, and the soil still adhered to the root surface was regarded as the ‘rhizosphere soil’. Therefore, the soil sample in this paper also can be defined as rhizosphere soil. For the readers to understand, we have changed root-zone to rhizosphere.

(Edwards et al. 2015)

Fig. S2. Sampling and collection of the rhizocompartments.

Roots are collected from rice plants and soil is shaken off the roots to leave ~1mm of soil around the roots. The ~1 mm of soil is washed off in PBS and kept as the rhizosphere compartment. The clean roots are then washed twice more to remove remaining soil and placed into clean PBS in a 50 mL Falcon tube. The rhizoplane microbes are extracted by sonicating the roots with the rhizosphere compartment removed. The sonicated roots are then placed in a new, clean Falcon tube and sonicated twice more, decanting the PBS in the tube between sonications and refilling with clean PBS. These roots are then kept for extracting the endospheric microbes.

References

1. Shi, W., M. Li, G. Wei, R. Tian, C. Li, B. Wang, R. Lin, C. Shi, X. Chi, B. Zhou & Z. Gao (2019) The occurrence of potato common scab correlates with the community composition and function of the geocaulosphere soil microbiome. *Microbiome*, 7, 14.

2. Edwards, J., C. Johnson, C. Santos-Medellin, E. Lurie, N. K. Podishetty, S. Bhatnagar, J. A. Eisen & V. Sundaresan (2015) Structure, variation, and assembly of the root-associated microbiomes of rice. *Proc Natl Acad Sci US A*, 112, E911-20.
3. Gremion, F., Chatzinotas, A., & Harms, H. (2003). Comparative 16S rDNA and 16S rRNA sequence analysis indicates that Actinobacteria might be a dominant part of the metabolically active bacteria in heavy metal-contaminated bulk and rhizosphere soil. *Environmental Microbiology*, 5(10), 896–907.
4. Lavecchia, A., Curci, M., Jangid, K., Whitman, W. B., Ricciuti, P., Pascazio, S., & Crecchio, C. (2015). Microbial 16S gene-based composition of a sorghum cropped rhizosphere soil under different fertilization managements. *Biology and Fertility of Soils*, 51(6), 661–672.

2. Questions and comments : Sometimes AMF in the soil respond to abiotic factors, rather than biotic factors. With the way this study was performed, it would be difficult to separate these factors because the plants are not evenly distributed throughout the study area.

Response 2: We appreciate the reviewer's insightful comments.

We mainly want to explore the diversity, structure and network interaction of AM fungi in the rhizosphere of *Stipa* species on the basis of its original succession distribution , and whether AM fungi are also changing with the change of *Stipa* populations on the whole regional scale. In this process, there are differences in *Stipa* populations (*Stipa* species), soil properties and climate. We only explore which factors play a greater role through correlation analysis, and the final conclusion also showed that it was of the combined outcome of multiple complicated ecological processes, but the influence of some factors will be more prominent.

In this study area, a large number of previous studies have determined the ecological adaptability and ecological geographical distribution of *Stipa* populations distributed in this region.

The uneven distribution of *Stipa* populations are also the result of long-term succession adaptation.

The distribution range of different *Stipa* spp. are inconsistent, with regional distribution characteristics (Fig.2). For example, *Stipa grandis* is the most widely distributed, and *Stipa purpurea* is not distributed in this study area. But the distribution of *Stipa* individuals in the corresponding *Stipa* steppe are even (Figure). Although the distribution of *Stipa* populations in the region is inconsistent, it is the dominant species of the grassland in this region. In each type of *Stipa* grassland, the *Stipa* individuals are almost a clump followed by a clump, almost in a state of uniform distribution, with only inconsistent coverage on different steppes (*Stipa* grassland map). Therefore, our sampling method (Fig. 1) can fully represent the status of AM fungal community in the rhizosphere of *Stipa* species in this type of *Stipa* grassland at this site. As for the widely distributed *Stipa* populations, we can not just select one sample site, but multi-sites sampling in the whole distribution area, and at each site we need to collect samples as shown in Fig. 1, so as to represent the AM fungal community in the rhizosphere of *Stipa* species. Similar sampling research methods are also in the following references. It may be that my initial positioning and name of the collected soil led to the emergence of this problem.

Fig.1. Map of sample locations, landscape and schematic diagram in arid and semiarid *Stipa* steppe.

Fig.2. Schematic diagram of alternative distribution of *Stipa* populations.

Arising from the North-South heat difference and the East-West southeast monsoon wind system (leading to different precipitation gradients), the grassland in this study displays the difference and gradient of temperature and humidity from southeast to northwest, as an attempt to differentiate meadow grassland (*S. baicalensis* population), typical grassland (*S. grandis* and *S. krylovii* population) and desert grassland (*S. breviflora*, *S. gobica*, *S. klemenzii* and *S. glareosa* population).

Meadow steppe

Typical steppe

Typical steppe

Desert steppe

References

1. Na X, Xu T, Li M, et al. Variations of bacterial community diversity within the rhizosphere of three phylogenetically related perennial shrub plant species across environmental gradients[J]. *Frontiers in microbiology*, 2018, 9: 709.

2. Zhao L, Zhang K, Sun X, et al. Dynamics of arbuscular mycorrhizal fungi and glomalin in the rhizosphere of *Gymnocarpus przewalskii* in Northwest Desert, China[J]. *Applied Soil Ecology*, 2022, 170: 104251.

3. Zhang B, Zhang J, Liu Y, et al. Co-occurrence patterns of soybean rhizosphere microbiome at a continental scale[J]. *Soil Biology and Biochemistry*, 2018, 118: 178-186.

4. Zhang B, Zhang J, Liu Y, et al. Biogeography and ecological processes affecting root-associated bacterial communities in soybean fields across China[J]. *Science of the Total Environment*, 2018, 627: 20-27.

3. Questions and comments : The bioinformatics could also be improved. I highly recommend using DADA2 for the bioinformatics analysis, it is the current gold standard for Illumina MiSeq data because it helps correct for over-inflated alpha diversity (Straub et al. 2020). QIIME2 makes this easy (Bolyen et al. 2018). A new taxonomic classifier might help as well (Curry et al. 2022).

Response 3: Thank you for suggesting that we have adopted and recalculated the alpha diversity (characterizing AMF richness, diversity and evenness) indicators needed in this paper. The relevant data processing description and the corresponding results and discussions also were modified.

Re-manuscript: line146-162, line 243-281, figure2, Table S3, Table S5, bioinformatics analysis and references, all of this made corresponding modifications.

4. Questions and comments :

Abstract

Line 17: Might what to reword as "Exactly how AMF ..." - sounds better reading it

Response: It has been revised. Re-manuscript: line 13

Line 19: What exactly is meant by "adaptive relationship between plants"

Response: We changed it to " In this study, the changes and distribution of the rhizosphere AMF community, and its associations between hosts," . Re-manuscript: line 14-15

Line 20: I'm not exactly sure what is meant by "zonal"

Response:

Adapt to climate, the distribution of vegetation appeared as belt is called zonal vegetation. The key factor of zonal vegetation is regional climate characteristics. For example, from coastal to inland, due to different water conditions, the vegetation types in the middle latitudes also appear forest → grassland → desert replacement, known as vegetation longitude zonality. The precipitation in this study area decreased gradually from east to west, and the changes of *Stipa* species with hydrothermal combination were also changing. In this process, *Stipa* species adapted to more drought will replace *Stipa* species distributed in humid areas and present zonal distribution in more arid areas. For example, a typical example: with the decrease of precipitation, the replacement order of *S. baicalensis* → *S. grandis* → *S. krylovii*. Therefore, *Stipa* species belongs to zonal vegetation.

For readers to understand easily, we changed 'seven zonal *Stipa* species' to Re-manuscript line 17 'seven *Stipa* species with spatial substitution distribution characteristics'.

Line 21: "abundant resources (83 species) of AMF were isolated" -- not accurate wording

Response: We appreciate the reviewer's insightful comments. This part was deleted because of the abstract length requirement. Then we will pay attention to correcting such problems.

Line 23: "quantity" - do you mean relative abundance?

Response: I mean the species (OTU) numbers, and changes in community structure include changes in relative abundance at different taxonomic levels.

Re-manuscript line: 19 We modified quantity to species numbers.

5. Questions and comments:

Introduction

Line 36: Use "account"

Line 37: Should be "Earth"

Line 37: I would avoid using "aroused", maybe use "attracted" instead

Response: At the request of another reviewer, the introduction needed to be shortened. Lines 36 and 37 of the original manuscript were shortened and merged into "Arid and semiarid grasslands have attracted wide attention, arising from the declining forage species, soil desertification and modifications to soil microbial communities due to global climate change (e.g., severe and frequent drought and heavy rainfall events) and anthropogenic activities (e.g., grazing) (1-4)."

6. Questions and comments:

Methods

Line 133: Do you mean "habitats"?

Response: Re-manuscript line 401, we modified "habitats" to "growth rhythm".

Line 209: Specify the sampling depth

Response: 9953 sequences

7. Questions and comments:

Figures

Figure 2: I don't think you need to Have the family and the table. I typically just report the genus for AMF, you could combine this with a figure of the alpha diversity or beta diversity ordination.

Response: We appreciate the reviewer's insightful comments. We have deleted excess parts other than genus for AMF, and this part was shown in Table 2, because the results of this part are closely related to Table S4 and put in one paragraph, so it was not combined with a figure of the alpha diversity or beta diversity ordination.

Figure 3: I don't think this figure is necessary.

Response: We accepted the suggestion and removed the relevant content.

Figure 4: Use the full name (e.g. Faith's PD) and define the acronym somewhere in the methods. It might be useful to report PERMANOVA results in this figure. I don't think you need the NMDS figure, just the Partial least square discriminant plot. Also, I don't think there is a need for all three alpha diversity metrics, maybe just pick the most interesting and put the rest in the supplemental information (describe all in the methods and results section, though).

Response: We appreciate the reviewer's insightful comments. Combined with the opinion of another reviewer and your suggestion, we removed the pd index and added the index of Pielou's evenness. In addition, we removed the NMDS analysis. Re-manuscript line 146-171.

8. Questions and comments: It may be helpful to number the plant species (with a key Table 1) instead of using abbreviations; the abbreviations are so similar they blend together.

Response 8: We have adopted the idea of renumbering the original plant species and modifying all the points covered in the text. According to the succession order of *Stipa* population in the study transect, the arabic numeral was added on the basis of the original number (extracted from Latin scientific name) to facilitate the distinction. Renumbered as follows, Sba→Sba1, Sgr→Sgr2, Skr→Skr3, Sbr→Sbr4, Stg→Stg5, Skl→Skl6 and Sgl→Sgl7.

Other changes:

1. The entire manuscript has been formatted into the requested layout of Microbiology Spectrum.

And we follow the styles of ASM journals: Introduction, Results, Discussion, then last Materials and Methods.

2. According to the format requirements of the journal, an importance description were modified.

“**IMPORTANCE** This study fills a gap in our understanding of the soil arbuscular mycorrhizal fungi community distribution, community composition changes and diversity of *Stipa* species along different *Stipa* population substitution distributions and their adaptive relationship; furthermore, the difference in the glomalin-related soil protein content in the rhizosphere of different *Stipa* species and its contribution to the grassland organic carbon pool were investigated. These findings provide a theoretical basis for the protection and utilization of regional biodiversity resources and sustainable ecosystem development.”.

3. We reduced and refined the introduction.

The SparCC model was used for network analysis and the associated method descriptions, results, discussions, and results were corrected. Re-manuscript line 182-197 and line 285-307.

4. According to the journal' s word count requirements for research articles, we deleted and simplified some redundant results and conclusion. For example, the specific description of Figure S2.

We tried our best to improve the manuscript and made a lot changes in the manuscript. We appreciate for Editors/Reviewers' warm work earnestly, and hope that the correction will meet with approval.

Once again, thank you very much for your comments and suggestions

Dear Reviewer:

Thank you for your letter and comments concerning our manuscript entitled “Changes of AM Fungi Community and Glomalin in the Rhizosphere Along the Distribution Gradients of Zonal *Stipa* Population Across the Arid and Semiarid Steppe” (Spectrum01489-22). I have received new suggestions and those comments are very detailed, professional and very helpful for revising and improving our paper, as well as the important guiding significance to our researches. We strongly cherish this opportunity that you gave us to further revise. Based on your comments and requests, we have studied comments carefully and have made correction which we hope meet with approval.

For editors and reviewers convenience, we have highlighted the changes in blue according to reviewer comments in the annotated version of the revised manuscript (Revision, changes marked). In addition, the manuscript has been polished in English, but there is no specific mark. I highlight the questions and comments in red. The main corrections in the paper and the responds to the reviewer’s comments are as flowing:

Reviewer #2:

1. Questions and comments:

Structure of manuscript: The entire manuscript has not been formatted into the requested layout of Microbiology Spectrum. Should follow the styles of ASM journals: Introduction, Results, Discussion, than last Materials and Methods. The English usage such as "inclined to decrease" is wrong. Please use a professional company or native speaker to polish the English of this manuscript.

Response 1: We have adjusted and modified the style of this paper as the requested layout of Microbiology Spectrum, including structure, pictures and titles. We appreciate the reviewer’s

insightful comments. We have polished the English language by professional company.

2. Questions and comments:

Introduction

(1) I feel the introduction too long. One and a half page is enough. Please be concise.

(2) Why *Stipa* plants are important to be involved in this study? What's the ecological importance of the plants themselves, not only the AMF. This need to state in the introduction. Similar for the glomalin. Why is it important to be included into this study?

Response 2(1): We have refined and reduced according to the recommendations, but this study needs to explain the alternative distribution of seven *Stipa* species, so the length is relatively not reduced to your required length. From the original 1069 words reduced to 770 words.

Response 2(2): This important part is reflected in the first paragraph of the introduction. Marked up manuscript line 41-45 and line 50-54 and line 79-81.

3. Questions and comments:

(1) "The region is characterized by continental arid and semi-arid monsoonal climate accompanied by abrupt changes of temperature in the middle temperate zone." "The climate is distributed in a belt, and gradually transits from semi humid to 1semi-arid and arid area from east to west." "The area is characterized by multi prevailing winds,with northwest wind prevailing in winter and southerly and southeast wind in summer." " *Stipa* spp.is the dominant species in the study steppe, and *Leymus chinensis* and *Filifolium sibiricum* are also commonly developed."The four sentences need to provide literature references to support these claims.

Response 3(1): "The region is characterized by continental arid and semi-arid monsoonal climate accompanied by abrupt changes of temperature in the middle temperate zone." and "The

climate is distributed in a belt, and gradually transits from semi humid to 1semi-arid and arid area from east to west."

Reference: Bai X, Zhao W, Wang J, & Ferreira CSS. 2021. Precipitation drives the floristic composition and diversity of temperate grasslands in China. *Global Ecology and Conservation* 32.

Hu Q, Pan F, Pan X, Zhang D, Li Q, Pan Z, & Wei Y. 2015. Spatial analysis of climate change in Inner Mongolia during 1961–2012, China. *Applied Geography* 60:254-260.

" The area is characterized by multi prevailing winds,with northwest wind prevailing in winter and southerly and southeast wind in summer."

Reference: Yin Y, Liu H, He S, Zhao F, Zhu J, Wang H, Liu G, & Wu X. 2011. Patterns of local and regional grain size distribution and their application to Holocene climate reconstruction in semi-arid Inner Mongolia, China. *Palaeogeography, Palaeoclimatology, Palaeoecology* 307:168-176.

" *Stipa* spp.is the dominant species in the study steppe, and *Leymus chinensis* and *Filifolium sibiricum* are also commonly developed."

Reference: Wang Q, Bao Y, Nan J, & Xu D. 2020. AM fungal diversity and its impact across three types of mid-temperate steppe in Inner Mongolia, China. *Mycorrhiza* 30:97-108.

Yin Y, Liu H, He S, Zhao F, Zhu J, Wang H, Liu G, & Wu X. 2011. Patterns of local and regional grain size distribution and their application to Holocene climate reconstruction in semi-arid Inner Mongolia, China. *Palaeogeography, Palaeoclimatology, Palaeoecology* 307:168-176.

(2) Microbial community diversity and richness have been determined, however community evenness is missing. It is important to measure community evenness, si that we can see if the change of diversity it majorly contributed by taxonomic richness or community evenness. I recommend to use Pielou's Evenness (for example, doi: 10.1007/s00253-022-12038-2; 10.1128/Spectrum.00229-21).

Response 3(2): We accepted the recommendation, calculated Pielou ' s Evenness, and described the results and discussions. According to the proposal of reviewer 1, we retained the shannon and chao1 index, and removed the Pd index. In addition, according to the suggestion of reviewer#1, the diversity index was recalculated by using DADA2. Marked up manuscript line 146-162.

(3) Why use CCA plots not RDA plots? I find the value of the CCA1 axis and CCA2 axis are both very low (<5%), while other studies always have CCA axes >20%. The author can try RDA and see if they can get a better result.

Response 3(3): Because in this study, the size of the first axis of Axis _ length was used to determine the selection of RDA or CCA in Decision Curve Analysis (DCA), the default CCA graph is greater

than or equal to 3.5 in this study.

(4) For co-occurrence network of microbial community, SparCC model is better than the models based on simpler algorithms such as Pearson, Spearman, and Kendall. I recommend to adopt SparCC to re-calculate the network as the method used in doi 10.1016/j.biortech.2022.127549, in order to make the results more reliable.

Response 3(4): Thank you for suggesting that we have adopted and recalculated relevant topological indices, drawn new graphs (FIG. 3), and modified relevant results and discussions. Marked up manuscript line 182-197 and line 285-307.

(5) The authors said "AMF-specific primers were designed based on the most variable segment of the small subunit (SSU) rRNA gene region." The authors mean that they designed the primers by themselves?

Response 3(5): I ' m sorry. It ' s the problem we described. We didn't design primers. We use the common primers in the references (Gao C, Kim Y-C, Zheng Y, Yang W, Chen L, Ji N-N, Wan S-Q, & Guo L-D. 2016. Increased precipitation, rather than warming, exerts a strong influence on arbuscular mycorrhizal fungal community in a semiarid steppe ecosystem. *Botany* 94:459-469.) . According to the comments given by the reviewer, we have deleted this sentence.

Other changes:

1. According to the format requirements of the journal, an importance description were modified. "IMPORTANCE This study fills a gap in our understanding of the soil arbuscular mycorrhizal fungi community distribution, community composition changes and diversity of *Stipa* species along different *Stipa* population substitution distributions and their adaptive relationship; furthermore, the difference in the glomalin-related soil protein content in the rhizosphere of different *Stipa* species

and its contribution to the grassland organic carbon pool were investigated. These findings provide a theoretical basis for the protection and utilization of regional biodiversity resources and sustainable ecosystem development.”.

2. Figure 3 (original manuscript) and related contents were removed according to the requirements of reviewer#1.

And the NMDS analysis of Figure 4d (original manuscript) was removed. Eventually merge images and α diversity into a re-manuscript of Figure 2.

The description of Family and Order in Figure 2(original manuscript) had been removed, and change into Table 2.

3. According to the succession order of *Stipa* population in the study transect, the arabic numeral was added on the basis of the original number (extracted from Latin scientific name) to facilitate the distinction. Renumbered as follows, Sba→Sba1, Sgr→Sgr2, Skr→Skr3, Sbr→Sbr4, Stg→Stg5, Skl→Skl6 and Sgl→Sgl7. (Requests from other reviewers)

4. According to the journal' s word count requirements for research articles, we deleted and simplified some redundant results and conclusion. For example, the specific description of Figure S2.

We tried our best to improve the manuscript and made a lot changes in the manuscript. We appreciate for Editors/Reviewers' warm work earnestly, and hope that the correction will meet with approval.

Once again, thank you very much for your comments and suggestions

Dear Editors and Reviewers:

Thank you for your letter and comments concerning our manuscript entitled “Changes of AM Fungi Community and Glomalin in the Rhizosphere Along the Distribution Gradients of Zonal *Stipa* Population Across the Arid and Semiarid Steppe” (Spectrum01489-22). I have received new suggestions and those comments are very detailed, professional and very helpful for revising and improving our paper, as well as the important guiding significance to our researches. We strongly cherish this opportunity that you gave us to further revise. Based on your comments and requests, we have studied comments carefully and have made correction which we hope meet with approval.

For editors and reviewers convenience, we have highlighted the changes in blue according to reviewer comments in the annotated version of the revised manuscript (Revision, changes marked). In addition, the manuscript has been polished in English, but there is no specific mark. I highlight the questions and comments in red. The main corrections in the paper and the responds to the reviewer’s comments are as flowing:

Reviewer #1:

1. Questions and comments: It’s not clear to me if these samples are bulk soil or rhizosphere soil.

Response 1: After shaking off the loose soil, we use a disposable brush to brush the soil still adhered to the rhizosheath until no more soil was brushed down, and collect the root-zone soil samples. We do not use PBS to wash root segment and collect soil by centrifugation. It may be more suitable to call it root-zone soil. We mainly refer to the definitions in the following articles (Edwards et al. 2015, Shi et al. 2019). Although *Stipa* has rhizosheath, the surrounding soil is not as loose as desert soil. In many related scientific studies (Gremion, Chatzinotas and Harms. 2003; Lavecchia et al. 2015),

loosing (shaking off) soil around easily removed roots is called ‘Bulk Soil’, and the soil still adhered to the root surface was regarded as the ‘rhizosphere soil’. Therefore, the soil sample in this paper also can be defined as rhizosphere soil. For the readers to understand, we have changed root-zone to rhizosphere.

(Edwards et al. 2015)

Fig. S2. Sampling and collection of the rhizocompartments.

Roots are collected from rice plants and soil is shaken off the roots to leave ~1mm of soil around the roots. The ~1 mm of soil is washed off in PBS and kept as the rhizosphere compartment. The clean roots are then washed twice more to remove remaining soil and placed into clean PBS in a 50 mL Falcon tube. The rhizoplane microbes are extracted by sonicating the roots with the rhizosphere compartment removed. The sonicated roots are then placed in a new, clean Falcon tube and sonicated twice more, decanting the PBS in the tube between sonications and refilling with clean PBS. These roots are then kept for extracting the endospheric microbes.

References

1. Shi, W., M. Li, G. Wei, R. Tian, C. Li, B. Wang, R. Lin, C. Shi, X. Chi, B. Zhou & Z. Gao (2019) The occurrence of potato common scab correlates with the community composition and function of the geocaulosphere soil microbiome. *Microbiome*, 7, 14.

2. Edwards, J., C. Johnson, C. Santos-Medellin, E. Lurie, N. K. Podishetty, S. Bhatnagar, J. A. Eisen & V. Sundaresan (2015) Structure, variation, and assembly of the root-associated microbiomes of rice. *Proc Natl Acad Sci US A*, 112, E911-20.
3. Gremion, F., Chatzinotas, A., & Harms, H. (2003). Comparative 16S rDNA and 16S rRNA sequence analysis indicates that Actinobacteria might be a dominant part of the metabolically active bacteria in heavy metal-contaminated bulk and rhizosphere soil. *Environmental Microbiology*, 5(10), 896–907.
4. Lavecchia, A., Curci, M., Jangid, K., Whitman, W. B., Ricciuti, P., Pascazio, S., & Crecchio, C. (2015). Microbial 16S gene-based composition of a sorghum cropped rhizosphere soil under different fertilization managements. *Biology and Fertility of Soils*, 51(6), 661–672.

2. Questions and comments : Sometimes AMF in the soil respond to abiotic factors, rather than biotic factors. With the way this study was performed, it would be difficult to separate these factors because the plants are not evenly distributed throughout the study area.

Response 2: We appreciate the reviewer's insightful comments.

We mainly want to explore the diversity, structure and network interaction of AM fungi in the rhizosphere of *Stipa* species on the basis of its original succession distribution , and whether AM fungi are also changing with the change of *Stipa* populations on the whole regional scale. In this process, there are differences in *Stipa* populations (*Stipa* species), soil properties and climate. We only explore which factors play a greater role through correlation analysis, and the final conclusion also showed that it was of the combined outcome of multiple complicated ecological processes, but the influence of some factors will be more prominent.

In this study area, a large number of previous studies have determined the ecological adaptability and ecological geographical distribution of *Stipa* populations distributed in this region.

The uneven distribution of *Stipa* populations are also the result of long-term succession adaptation.

The distribution range of different *Stipa* spp. are inconsistent, with regional distribution characteristics (Fig.2). For example, *Stipa grandis* is the most widely distributed, and *Stipa purpurea* is not distributed in this study area. But the distribution of *Stipa* individuals in the corresponding *Stipa* steppe are even (Figure). Although the distribution of *Stipa* populations in the region is inconsistent, it is the dominant species of the grassland in this region. In each type of *Stipa* grassland, the *Stipa* individuals are almost a clump followed by a clump, almost in a state of uniform distribution, with only inconsistent coverage on different steppes (*Stipa* grassland map). Therefore, our sampling method (Fig. 1) can fully represent the status of AM fungal community in the rhizosphere of *Stipa* species in this type of *Stipa* grassland at this site. As for the widely distributed *Stipa* populations, we can not just select one sample site, but multi-sites sampling in the whole distribution area, and at each site we need to collect samples as shown in Fig. 1, so as to represent the AM fungal community in the rhizosphere of *Stipa* species. Similar sampling research methods are also in the following references. It may be that my initial positioning and name of the collected soil led to the emergence of this problem.

Fig.1. Map of sample locations, landscape and schematic diagram in arid and semiarid *Stipa* steppe.

Fig.2. Schematic diagram of alternative distribution of *Stipa* populations.

Arising from the North-South heat difference and the East-West southeast monsoon wind system (leading to different precipitation gradients), the grassland in this study displays the difference and gradient of temperature and humidity from southeast to northwest, as an attempt to differentiate meadow grassland (*S. baicalensis* population), typical grassland (*S. grandis* and *S. krylovii* population) and desert grassland (*S. breviflora*, *S. gobica*, *S. klemenzii* and *S. glareosa* population).

Meadow steppe

Typical steppe

Typical steppe

Desert steppe

References

1. Na X, Xu T, Li M, et al. Variations of bacterial community diversity within the rhizosphere of three phylogenetically related perennial shrub plant species across environmental gradients[J]. *Frontiers in microbiology*, 2018, 9: 709.

2. Zhao L, Zhang K, Sun X, et al. Dynamics of arbuscular mycorrhizal fungi and glomalin in the rhizosphere of *Gymnocarpus przewalskii* in Northwest Desert, China[J]. *Applied Soil Ecology*, 2022, 170: 104251.

3. Zhang B, Zhang J, Liu Y, et al. Co-occurrence patterns of soybean rhizosphere microbiome at a continental scale[J]. *Soil Biology and Biochemistry*, 2018, 118: 178-186.

4. Zhang B, Zhang J, Liu Y, et al. Biogeography and ecological processes affecting root-associated bacterial communities in soybean fields across China[J]. *Science of the Total Environment*, 2018, 627: 20-27.

3. Questions and comments : The bioinformatics could also be improved. I highly recommend using DADA2 for the bioinformatics analysis, it is the current gold standard for Illumina MiSeq data because it helps correct for over-inflated alpha diversity (Straub et al. 2020). QIIME2 makes this easy (Bolyen et al. 2018). A new taxonomic classifier might help as well (Curry et al. 2022).

Response 3: Thank you for suggesting that we have adopted and recalculated the alpha diversity (characterizing AMF richness, diversity and evenness) indicators needed in this paper. The relevant data processing description and the corresponding results and discussions also were modified.

Re-manuscript: line146-162, line 243-281, figure2, Table S3, Table S5, bioinformatics analysis and references, all of this made corresponding modifications.

4. Questions and comments :

Abstract

Line 17: Might what to reword as "Exactly how AMF ..." - sounds better reading it

Response: It has been revised. Re-manuscript: line 13

Line 19: What exactly is meant by "adaptive relationship between plants"

Response: We changed it to " In this study, the changes and distribution of the rhizosphere AMF community, and its associations between hosts," . Re-manuscript: line 14-15

Line 20: I'm not exactly sure what is meant by "zonal"

Response:

Adapt to climate, the distribution of vegetation appeared as belt is called zonal vegetation. The key factor of zonal vegetation is regional climate characteristics. For example, from coastal to inland, due to different water conditions, the vegetation types in the middle latitudes also appear forest → grassland → desert replacement, known as vegetation longitude zonality. The precipitation in this study area decreased gradually from east to west, and the changes of *Stipa* species with hydrothermal combination were also changing. In this process, *Stipa* species adapted to more drought will replace *Stipa* species distributed in humid areas and present zonal distribution in more arid areas. For example, a typical example: with the decrease of precipitation, the replacement order of *S. baicalensis* → *S. grandis* → *S. krylovii*. Therefore, *Stipa* species belongs to zonal vegetation.

For readers to understand easily, we changed 'seven zonal *Stipa* species' to Re-manuscript line 17 'seven *Stipa* species with spatial substitution distribution characteristics'.

Line 21: "abundant resources (83 species) of AMF were isolated" -- not accurate wording

Response: We appreciate the reviewer's insightful comments. This part was deleted because of the abstract length requirement. Then we will pay attention to correcting such problems.

Line 23: "quantity" - do you mean relative abundance?

Response: I mean the species (OTU) numbers, and changes in community structure include changes in relative abundance at different taxonomic levels.

Re-manuscript line: 19 We modified quantity to species numbers.

5. Questions and comments:

Introduction

Line 36: Use "account"

Line 37: Should be "Earth"

Line 37: I would avoid using "aroused", maybe use "attracted" instead

Response: At the request of another reviewer, the introduction needed to be shortened. Lines 36 and 37 of the original manuscript were shortened and merged into "Arid and semiarid grasslands have attracted wide attention, arising from the declining forage species, soil desertification and modifications to soil microbial communities due to global climate change (e.g., severe and frequent drought and heavy rainfall events) and anthropogenic activities (e.g., grazing) (1-4)."

6. Questions and comments:

Methods

Line 133: Do you mean "habitats"?

Response: Re-manuscript line 401, we modified "habitats" to "growth rhythm".

Line 209: Specify the sampling depth

Response: 9953 sequences

7. Questions and comments:

Figures

Figure 2: I don't think you need to Have the family and the table. I typically just report the genus for AMF, you could combine this with a figure of the alpha diversity or beta diversity ordination.

Response: We appreciate the reviewer's insightful comments. We have deleted excess parts other than genus for AMF, and this part was shown in Table 2, because the results of this part are closely related to Table S4 and put in one paragraph, so it was not combined with a figure of the alpha diversity or beta diversity ordination.

Figure 3: I don't think this figure is necessary.

Response: We accepted the suggestion and removed the relevant content.

Figure 4: Use the full name (e.g. Faith's PD) and define the acronym somewhere in the methods. It might be useful to report PERMANOVA results in this figure. I don't think you need the NMDS figure, just the Partial least square discriminant plot. Also, I don't think there is a need for all three alpha diversity metrics, maybe just pick the most interesting and put the rest in the supplemental information (describe all in the methods and results section, though).

Response: We appreciate the reviewer's insightful comments. Combined with the opinion of another reviewer and your suggestion, we removed the pd index and added the index of Pielou's evenness. In addition, we removed the NMDS analysis. Re-manuscript line 146-171.

8. Questions and comments: It may be helpful to number the plant species (with a key Table 1) instead of using abbreviations; the abbreviations are so similar they blend together.

Response 8: We have adopted the idea of renumbering the original plant species and modifying all the points covered in the text. According to the succession order of *Stipa* population in the study transect, the arabic numeral was added on the basis of the original number (extracted from Latin scientific name) to facilitate the distinction. Renumbered as follows, Sba→Sba1, Sgr→Sgr2, Skr→Skr3, Sbr→Sbr4, Stg→Stg5, Skl→Skl6 and Sgl→Sgl7.

Other changes:

1. The entire manuscript has been formatted into the requested layout of Microbiology Spectrum.

And we follow the styles of ASM journals: Introduction, Results, Discussion, than last Materials and Methods.

2. According to the format requirements of the journal, an importance description were modified.

“**IMPORTANCE** This study fills a gap in our understanding of the soil arbuscular mycorrhizal fungi community distribution, community composition changes and diversity of *Stipa* species along different *Stipa* population substitution distributions and their adaptive relationship; furthermore, the difference in the glomalin-related soil protein content in the rhizosphere of different *Stipa* species and its contribution to the grassland organic carbon pool were investigated. These findings provide a theoretical basis for the protection and utilization of regional biodiversity resources and sustainable ecosystem development.”.

3. We reduced and refined the introduction.

The SparCC model was used for network analysis and the associated method descriptions, results, discussions, and results were corrected. Re-manuscript line 182-197 and line 285-307.

4. According to the journal' s word count requirements for research articles, we deleted and simplified some redundant results and conclusion. For example, the specific description of Figure S2.

Reviewer #2:

1. Questions and comments:

Structure of manuscript: The entire manuscript has not been formatted into the requested layout of Microbiology Spectrum. Should follow the styles of ASM journals: Introduction, Results, Discussion, than last Materials and Methods. The English usage such as "inclined to decrease" is wrong. Please use a professional company or native speaker to polish the English of this manuscript.

Response 1 : We have adjusted and modified the style of this paper as the requested layout of

Microbiology Spectrum, including structure, pictures and titles. We appreciate the reviewer's insightful comments. We have polished the English language by professional company.

2. Questions and comments:

Introduction

(1) I feel the introduction too long. One and a half page is enough. Please be concise.

(2) Why *Stipa* plants are important to be involved in this study? What's the ecological importance of the plants themselves, not only the AMF. This need to state in the introduction. Similar for the glomalin. Why is it important to be included into this study?

Response 2(1): We have refined and reduced according to the recommendations, but this study needs to explain the alternative distribution of seven *Stipa* species, so the length is relatively not reduced to your required length. From the original 1069 words reduced to 770 words.

Response 2(2): This important part is reflected in the first paragraph of the introduction. Marked up manuscript line 41-45 and line 50-54 and line 79-81.

3. Questions and comments:

(1) "The region is characterized by continental arid and semi-arid monsoonal climate accompanied by abrupt changes of temperature in the middle temperate zone." "The climate is distributed in a belt, and gradually transits from semi humid to 1semi-arid and arid area from east to west." "The area is characterized by multi prevailing winds,with northwest wind prevailing in winter and southerly and southeast wind in summer." " *Stipa* spp.is the dominant species in the study steppe, and *Leymus chinensis* and *Filifolium sibiricum* are also commonly developed."The four sentences need to provide literature references to support these claims.

Response 3(1): "The region is characterized by continental arid and semi-arid monsoonal climate accompanied by abrupt changes of temperature in the middle temperate zone." and "The climate is distributed in a belt, and gradually transits from semi humid to 1semi-arid and arid area from east to west."

Reference: Bai X, Zhao W, Wang J, & Ferreira CSS. 2021. Precipitation drives the floristic composition and diversity of temperate grasslands in China. *Global Ecology and Conservation* 32.

Hu Q, Pan F, Pan X, Zhang D, Li Q, Pan Z, & Wei Y. 2015. Spatial analysis of climate change in Inner Mongolia during 1961–2012, China. *Applied Geography* 60:254-260.

" The area is characterized by multi prevailing winds,with northwest wind prevailing in winter and southerly and southeast wind in summer."

Reference: Yin Y, Liu H, He S, Zhao F, Zhu J, Wang H, Liu G, & Wu X. 2011. Patterns of local and regional grain size distribution and their application to Holocene climate reconstruction in semi-arid Inner Mongolia, China. *Palaeogeography, Palaeoclimatology, Palaeoecology* 307:168-176.

" *Stipa* spp.is the dominant species in the study steppe, and *Leymus chinensis* and *Filifolium sibiricum* are also commonly developed."

Reference: Wang Q, Bao Y, Nan J, & Xu D. 2020. AM fungal diversity and its impact across three types of mid-temperate steppe in Inner Mongolia, China. *Mycorrhiza* 30:97-108.

Yin Y, Liu H, He S, Zhao F, Zhu J, Wang H, Liu G, & Wu X. 2011. Patterns of local and regional grain size distribution and their application to Holocene climate reconstruction in semi-arid Inner Mongolia, China. *Palaeogeography, Palaeoclimatology, Palaeoecology* 307:168-176.

(2) Microbial community diversity and richness have been determined, however community evenness is missing. It is important to measure community evenness, si that we can see if the change of diversity it majorly contributed by taxonomic richness or community evenness. I recommend to use Pielou's Evenness (for example, doi: 10.1007/s00253-022-12038-2; 10.1128/Spectrum.00229-21).

Response 3(2): We accepted the recommendation, calculated Pielou ' s Evenness, and described the results and discussions. According to the proposal of reviewer 1, we retained the shannon and chao1 index, and removed the Pd index. In addition, according to the suggestion of reviewer#1, the diversity index was recalculated by using DADA2. Marked up manuscript line 146-162.

(3) Why use CCA plots not RDA plots? I find the value of the CCA1 axis and CCA2 axis are both very low (<5%), while other studies always have CCA axes >20%. The author can try RDA and see if they can get a better result.

Response 3(3): Because in this study, the size of the first axis of Axis _ length was used to determine

the selection of RDA or CCA in Decision Curve Analysis (DCA), the default CCA graph is greater than or equal to 3.5 in this study.

(4) For co-occurrence network of microbial community, SparCC model is better than the models based on simpler algorithms such as Pearson, Spearman, and Kendall. I recommend to adopt SparCC to re-calculate the network as the method used in doi 10.1016/j.biortech.2022.127549, in order to make the results more reliable.

Response 3(4): Thank you for suggesting that we have adopted and recalculated relevant topological indices, drawn new graphs (FIG. 3), and modified relevant results and discussions. Marked up manuscript line 182-197 and line 285-307.

(5) The authors said "AMF-specific primers were designed based on the most variable segment of the small subunit (SSU) rRNA gene region." The authors mean that they designed the primers by themselves?

Response 3(5): I ' m sorry. It ' s the problem we described. We didn't design primers. We use the common primers in the references (Gao C, Kim Y-C, Zheng Y, Yang W, Chen L, Ji N-N, Wan S-Q, & Guo L-D. 2016. Increased precipitation, rather than warming, exerts a strong influence on arbuscular mycorrhizal fungal community in a semiarid steppe ecosystem. *Botany* 94:459-469.) . According to the comments given by the reviewer, we have deleted this sentence.

Other changes:

1. According to the format requirements of the journal, an importance description were modified. **“IMPORTANCE** This study fills a gap in our understanding of the soil arbuscular mycorrhizal fungi community distribution, community composition changes and diversity of *Stipa* species along different *Stipa* population substitution distributions and their adaptive relationship; furthermore, the

difference in the glomalin-related soil protein content in the rhizosphere of different *Stipa* species and its contribution to the grassland organic carbon pool were investigated. These findings provide a theoretical basis for the protection and utilization of regional biodiversity resources and sustainable ecosystem development.”.

2. Figure 3 (original manuscript) and related contents were removed according to the requirements of reviewer#1.

And the NMDS analysis of Figure 4d (original manuscript) was removed. Eventually merge images and α diversity into a re-manuscript of Figure 2.

The description of Family and Order in Figure 2(original manuscript) had been removed, and change into Table 2.

3. According to the succession order of *Stipa* population in the study transect, the arabic numeral was added on the basis of the original number (extracted from Latin scientific name) to facilitate the distinction. Renumbered as follows, Sba→Sba1, Sgr→Sgr2, Skr→Skr3, Sbr→Sbr4, Stg→Stg5, Skl→Skl6 and Sgl→Sgl7. (Requests from other reviewers)

4. According to the journal' s word count requirements for research articles, we deleted and simplified some redundant results and conclusion. For example, the specific description of Figure S2.

We tried our best to improve the manuscript and made a lot changes in the manuscript. We appreciate for Editors/Reviewers' warm work earnestly, and hope that the correction will meet with approval.

Once again, thank you very much for your comments and suggestions

September 20, 2022

Dr. Xiaodan Ma
Inner Mongolia University
School of Life Sciences, Inner Mongolia University
Hohhot
China

Re: Spectrum01489-22R1 (Changes of AM Fungi Community and Glomalin in the Rhizosphere Along the Distribution Gradients of Zonal Stipa Population Across the Arid and Semiarid Steppe.)

Dear Dr. Xiaodan Ma:

Your manuscript has been accepted, and I am forwarding it to the ASM Journals Department for publication. You will be notified when your proofs are ready to be viewed. Please ensure all data is released per your data availability statement.

Sincerely,

Christina Cuomo
Editor, Microbiology Spectrum
